# IpaA reveals distinct modes of vinculin activation during *Shigella* invasion and cell-matrix adhesion

Benjamin Cocom-Chan[1,2,3], Hamed Khakzad[1,2,3,4], Mahamadou Konate[1,2,3], Daniel Isui Aguilar[5,6,7,8], Chakir Bello[5,6,7,8], Cesar Valencia-Gallardo[5,6,7,8], Yosra Zarrouk[1,2,3], Jacques Fattaccioli[9,10], Alain Mauviel[11,12,13], Delphine Javelaud[11,12,13], Guy Tran Van Nhieu[1,2,3,5,6,7,8]

Vinculin is a cytoskeletal linker strengthening cell adhesion. The *Shigella* IpaA invasion effector binds to vinculin to promote vinculin supra-activation associated with head-domain–mediated oligomerization. Our study investigates the impact of mutations of vinculin D1D2 subdomains' residues predicted to interact with IpaA VBS3. These mutations affected the rate of D1D2 trimer formation with distinct effects on monomer disappearance, consistent with structural modeling of a *closed* and *open* D1D2 conformer induced by IpaA. Notably, mutations targeting the closed D1D2 conformer significantly reduced *Shigella* invasion of host cells as opposed to mutations targeting the open D1D2 conformer and later stages of vinculin head-domain oligomerization. In contrast, all mutations affected the formation of focal adhesions (FAs), supporting the involvement of vinculin supra-activation in this process. Our findings suggest that IpaA-induced vinculin supra-activation primarily reinforces matrix adhesion in infected cells, rather than promoting bacterial invasion. Consistently, shear stress studies pointed to a key role for IpaA-induced vinculin supra-activation in accelerating and strengthening cell-matrix adhesion.

## Introduction

Vinculin is a cytoskeletal linker of integrin-mediated matrix adhesions and cadherin-based cell-cell junctions (Goldman, 2016; Bays & DeMali, 2017). It plays an important role in cell adhesion processes, motility and development and its functional deficiency is associated with major diseases including cancer and cardiomyopathies (Peng et al, 2011). Vinculin associates with a number of ligands, including focal adhesion and intercellular junction components, lipids, signaling proteins as well as proteins regulating the organization and dynamics of the actin cytoskeleton (Goldman, 2016; Bays & DeMali, 2017). The role of vinculin in integrin-mediated adhesion has been particularly studied (Parsons et al, 2010; Atherton et al, 2016). Vinculin is recruited at focal adhesions and reinforces the link between integrins and the actin cytoskeleton. The extent of vinculin recruitment determines the growth and maturation of focal adhesions, associated with the scaffolding of adhesion components and mechanotransduction linked to the actomyosin contraction (Parsons et al, 2010; Atherton et al, 2016). Whereas the precise mechanisms leading to vinculin activation involving combinatorial stimulation and recruitment at focal adhesions in cells are not fully understood, in vitro biomimetic mechanical and structure-function studies have enlightened major aspects of the role of vinculin in cell adhesion (Yan et al, 2015).

Vinculin is classically described as a three-domain protein containing an amino-terminal globular head domain (Vh), a flexible linker domain, and a F-actin-binding tail domain (Vt). Vh contains three subdomains D1-D3 and a small subdomain D4. With the exception of D4 containing a single helix bundle, each D1-D3 subdomain corresponds to a conserved repeat consisting of two four/five helix bundles connected by a long alpha-helix (Fig 1A; Goldman, 2016). At the inactive state, vinculin is maintained folded by intramolecular interactions between Vh and Vt. All ligands activating vinculin contain a vinculin-binding site (VBS), corresponding to 20–25 residues structured into an amphipathic α-helix that interacts with the first helix-bundle of D1 (Gingras et al, 2005). Insertion of the activating VBS in the D1 first helix bundle leads to the reorganization of the bundle and destabilizes the interaction between D1 and Vt (Izard et al, 2004). More than 70 VBSs have been identified in the various vinculin ligands, often containing multiple VBSs (Kluger et al, 2020). These VBSs, however, are often buried into helix bundles and their exposure regulates vinculin activation

[1]Team "Ca[2+] Signaling and Microbial Infections", I2BC, Gif-sur-Yvette, France [2]Institut National de la Santé et de la Recherche Médicale U1280, Gif-sur-Yvette, France [3]Centre National de la Recherche Scientifique UMR9198, Gif-sur-Yvette, France [4]Université de Lorraine, CNRS, Inria, LORIA, Nancy, France [5]Equipe Communication Intercellulaire et Infections Microbiennes, Centre de Recherche Interdisciplinaire en Biologie (CIRB), Collège de France, Paris, France [6]Institut National de la Santé et de la Recherche Médicale U1050, Paris, France [7]Centre National de la Recherche Scientifique UMR7241, Paris, France [8]MEMOLIFE Laboratory of Excellence and Paris Science Lettre, Paris, France [9]PASTEUR, Département de Chimie, École Normale Supérieure, PSL University, Sorbonne Université, CNRS, Paris, France [10]Institut Pierre-Gilles de Gennes pour la Microfluidique, Paris, France [11]Institut Curie, PSL Research University, INSERM U1021, CNRS UMR3347, Team "TGF-ß and Oncogenesis", Equipe Labellisée LIGUE 2016, Orsay, France [12]Université Paris-Sud, Orsay, France [13]Centre National de la Recherche Scientifique UMR 3347, Orsay, France

Correspondence: guy.tranvannhieu@i2bc.paris-saclay.fr

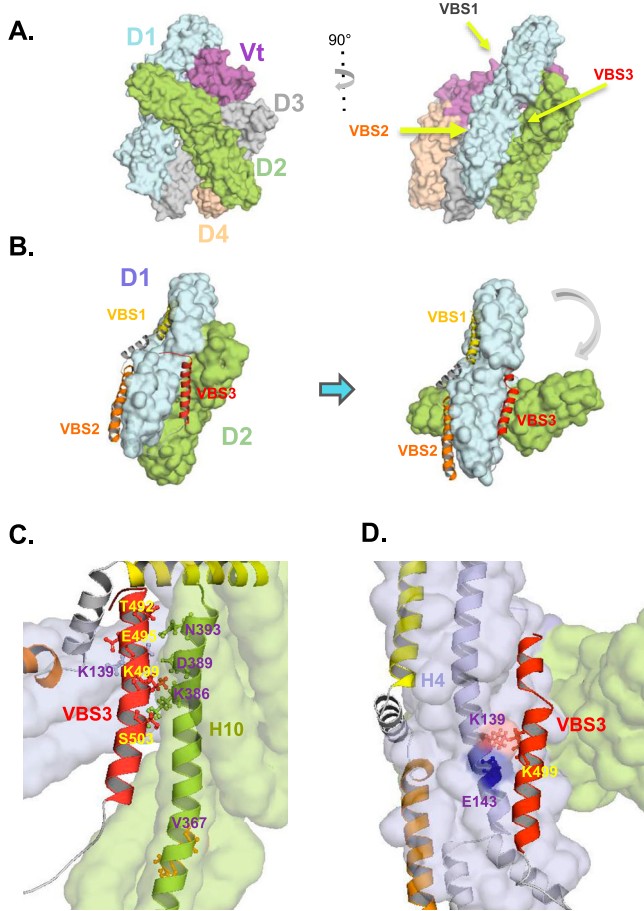

**Figure 1. Design of D1D2 mutations targeting IpaA VBS3 contact sites.**
**(A)** Surface structure of full-length human vinculin from the crystal structure resolved by Bakolitsa et al (1999). The vinculin subdomains are indicated in different colors. Right: The arrows point at binding of IpaA VBS1 to the D1 first bundle, IpaA VBS2 to the second D1 bundle, IpaA VBS3 binding to the D1D2 interface and D2 second bundle. **(B, C, D)** cross-linking mass spectrometry-based modeling-based models of 1:1 complexes of D1D2 and the IpaA domain containing all three IpaA VBS1–3 (aVBD) (Valencia-Gallardo et al, 2023). The vinculin D1 and D2 are shown as pale cyan and limon green surface structures, respectively. IpaA VBSs are shown as ribbon structures. **(B)** Left: aVBD binds to D1D2 that adopts a conformation similar to that of apoD1D2 (closed conformer). Right: binding of IpaA VBS3 to the D1D2 interface induces a major conformational change (open conformer) with a 30° tilt in the relative orientation of the D1 and D2 major axis. **(C, D)** higher magnifications showing the IpaA VBS3(red) interface with D1 D2 in the closed (C) and open (D) conformer. The residues potentially involved in polar interactions via their side chains are indicated. In the open conformer, IpaA K499 may interact with E143 or clash with K139 on vinculin D1.

(Kluger et al, 2020). Force-induced stretching of a vinculin ligand such as talin, acting as mechanosensor, provides a means to expose the VBSs during integrin-mediated adhesion (Sun et al, 2016; Goult et al, 2021). Talin binds to F-actin via at least two sites in its rod domain and to integrin cytoplasmic domains via its amino-terminal FERM domain. During mechanotransduction, talin stretching by the actomyosin contraction exposes VBSs that are buried in helix bundles of the rod domain in the native state (Yan et al, 2015). Exposed talin VBSs in turn bind to vinculin and relieve the intramolecular interactions between the vinculin head (Vh) and tail (Vt) domains, unveiling the F-actin-binding site in Vt (Goult et al, 2021).

Because talin contains 11 VBSs in helix bundles unfolding at different force amplitudes, the stretching force-dependent interaction between talin and vinculin serves as a mechanism to strengthen actin cytoskeletal anchorage as a function of substrate stiffness (Yan et al, 2015; Yao et al, 2016). In vitro, vinculin interaction with a VBS is sufficient to promote its opening and interaction with F-actin. Vinculin activation in cells, however, may result from a combinatorial stimulus including interaction with F-actin, phosphatidylinositol (4,5)-biphosphate (PIP2) or phosphorylation (Auernheimer et al, 2015; Izard & Brown, 2016).

Intracellular bacterial pathogens such as *Chlamydia*, *Rickettsia*, and *Shigella* express ligands diverting vinculin functions to promote virulence (Park et al, 2011a; Thwaites et al, 2015; Valencia-Gallardo et al, 2015). Among these, the *Shigella* type III effector IpaA was shown to target vinculin via three VBSs present at its carboxyterminal domain (Valencia-Gallardo et al, 2015). Unlike other host cell endogenous VBSs, IpaA VBSs are not buried into helix bundles and therefore likely act in concert to promote bacterial invasion. IpaA VBS1 and VBS2 bind to the first and second bundles of D1, respectively, conferring binding to vinculin with a very high affinity and the IpaA property to act as a "super-mimic" of endogenous activating VBSs (Fig 1A and B; Izard et al, 2006; Tran Van Nhieu & Izard, 2007). The role of IpaA VBS3 appears more complex and likely underlines the role of IpaA in different processes during bacterial invasion. In addition to vinculin, IpaA VBS3 also binds to talin and may stabilize a partially stretched talin conformer present in filopodial adhesions, thereby favoring bacterial capture by filopodia at the initial stages of the bacterial invasion process (Park et al, 2011b; Valencia-Gallardo et al, 2019). IpaA VBS3 was also shown to bind to the vinculin D2 subdomain when IpaA VBSs 1–2 (aVBS1-2) are bound to D1 (Valencia-Gallardo et al, 2023). Binding of IpaA VBS1-3 (aVBD) to D1D2 triggers major conformational changes, coined "supra-activation," leading to D1D2 homo-oligomerization via the D1D2 head subdomains and the formation of D1D2 trimers (Valencia-Gallardo et al, 2023). IpaA-induced vinculin supra-activation enables invasive *Shigella* to promote strong adhesion in the absence of mechanotransduction (Valencia-Gallardo et al, 2023). In addition, analysis of a vinculin cysteine-clamp variant (HV-CC), deficient for vinculin supra-activation but still proficient for canonical activation, suggests that vinculin head domain oligomerization is required for vinculin-dependent actin bundling and the maturation of focal adhesions into large adhesion structures (Valencia-Gallardo et al, 2023).

Seminal studies based on rotary shadow electron microscopy analysis reported the formation of vinculin mediated by Vh-Vh and Vt-Vt interactions in vitro (Molony & Burridge, 1985). Ever since studies on vinculin oligomerization have essentially focused on Vt-Vt interactions. Phosphatidylinositol (4, 5) bisphosphate (PIP2) binding to vinculin was shown to promote vinculin oligomerization, and PIP2-binding deficient vinculin showed defects in the organization of the actin cytoskeleton and increased turn-over of focal adhesions (Bakolitsa et al, 1999; Chinthalapudi et al, 2014). Binding of F-actin was also reported to induce the formation of vinculin tail dimers, likely different than those induced by PIP2, and mutants in the Vt C-terminal hairpin responsible for F-actin-induced dimerization showed defects in actin bundling associated with a decrease in size and a number of focal adhesions (Bakolitsa et al, 1999; Johnson & Craig,

2000). Vinculin establishes catch bonds with a significantly increased lifetime when the force is applied towards the pointed end of actin filaments, consistent with the polarity of actomyosin contraction (Huang et al, 2017). However, whereas mechanotransduction triggers vinculin recruitment associated with the maturation, enlarging of focal adhesions and actin bundling, how Vt-mediated vinculin oligomerization may be regulated by actomyosin contraction remains unclear (Thompson et al, 2013). Vinculin head-domain (Vh)–mediated oligomerization could provide a force-dependent mechanism, because vinculin was reported to act as a mechanosensor, with its head domain undergoing conformational changes under applied force, showing increased binding to ligand such as MAPK1 (Garakani et al, 2017). Here, we report the effects of mutation in vinculin D1D2 altering the formation of trimers induced by IpaA. We identified D1D2 polar residues predicted to contact IpaA VBS3 and showed their involvement in IpaA-induced trimerization. We show that these mutations affect the formation of focal adhesions providing evidence that Vh-Vh–mediated vinculin oligomerization similar to that induced by *Shigella* IpaA may also occur during the maturation of cell adhesion structures.

# Results

### Design of mutations in D1D2 affecting IpaA-induced trimer formation

Structural models indicate that aVBD triggers a 30% angle displacement of the major axis of the D1 and D2 subdomains relative to the apo D1D2 or D1D2 in complex with IpaA VBS1-2 only (Fig 1B; Valencia-Gallardo et al, 2023). As opposed to the *closed* D1D2 conformer, this *open* D1D2 conformer is associated with D1D2-mediated trimerization of vinculin (Valencia-Gallardo et al, 2023). We hypothesized that IpaA VBS3 induced allosteric changes leading to D1D2 trimerization and that mutations in D1D2 residues interfacing IpaA VBS3 should affect trimer formation. We, therefore, scrutinized the interface residues between IpaA VBS3 and D1D2 in the D1D2:aVBD complex.

As shown in Fig 1C, in the closed D1D2:aVBD complex, IpaA VBS3 mainly interacts with the H10 helix of D2. A set of putative polar interactions and salt bridges can be identified, where IpaA residues T492, E495, K499, and S503 interact with vinculin residues N393, K139, D389, and K386, respectively (Fig 1C). In the open complex, IpaA VBS3 mainly interacts with the H4 helix of D1, where IpaA K499 interacts with vinculin E143 (Fig 1D). Of note, in this open complex, an electrostatic clash between vinculin K139 and IpaA K499 may contribute to the dynamics of the IpaA VBS3 during its interaction with different allosteric conformers leading to D1D2 oligomerization (Fig 1D). All identified vinculin residues were substituted for a charged residue to introduce a charge inversion or disrupt polar interactions using site-directed mutagenesis (see the Materials and Methods section; Table 1). In this rationale, mutations affecting IpaA VBS3's interface with the close D1D2 conformer are expected to alter its initial docking of IpaA VBS3 on D1D2, whereas the E143K charge inversion would destabilize the open conformer and alter subsequent allosteric changes leading to trimer formation.

Independent of the aVBD-D1D2 interface, the hidden Markov model-based algorithm MARCOIL predicts the presence of a coiled-coil domain in the H10 helix of D2 between vinculin residues 348–393, buried into the D2 helix bundle and adjacent to the IpaA VBS3 interaction sites in the close D1D2 conformer (Delorenzi & Speed, 2002; Fig S1). This putative coiled-coil domain contains the classical a-g heptad sequence at residues 367–373, with the V367 and A373 corresponding to the "a" and "d" hydrophobic residues, respectively, presumed to intersperse their non-polar side chains at the $\alpha$-helices interfaces during oligomerization. To test the role of this domain in D1D2 trimerization, we introduced the V367D substitution predicted to disrupt supercoiled helix packing (Fig S1A–D).

### Quantitative analysis of D1D2 trimer formation using CN-PAGE

In previous works, we showed that aVBD induced the formation of D1D2:aVBD 1:1 and dimeric complexes and trimeric D1D2 complexes rapidly associating and dissociating an aVBD molecule in SEC-MALS (Size Exclusion Chromatography-Multi-Angle Light Scattering) experiments (Valencia-Gallardo et al, 2023). Here, we studied the effects of increasing aVBD molar ratio using SEC and CN (Clear Native)-PAGE. As shown in Fig 2, at a D1D2:aVBD 1:1 M ratio, the major peaks with similar amplitude corresponded to the 1:0 and likely the D1D2 trimeric complexes (Fig 2A, peaks A and C). At a D1D2:aVBD 1:1.5 M ratio, the D1D2 trimeric complexes represented the major peak (Fig 2A, peak A). Upon increasing of IpaA molar ratio, a major shifted band was observed in CN-PAGE with apo D1D2 rapidly disappearing and the formation of intermediate shifted bands (Fig 2B). The similarity between the distribution of the bands with increasing molar ratio of aVBD in CN-PAGE and the SEC peaks suggested that the A and A' peaks corresponded to trimeric D1D2 complexes. To confirm this, we dissected the A' band form CN-PAGE and analyzed it in a second dimension using regular SDS–PAGE, followed by Coomassie staining and scanning densitometry (see the Materials and Methods section). As shown in Fig 2C and D, the relative amounts of D1D2 and AVBD in A' indicated a D1D2:aVBD integrated density ratio of 10.1 ± 0.36 corresponding to a molar ratio of 3.2 ± 0.09, consistent with a mixture of D1D2:aVBD 3:0 and 3:1 complexes inferred from the SEC-MALS analysis (Valencia-Gallardo et al, 2023).

We used scanning densitometry to determine the rates of D1D2 trimer formation and D1D2 monomer disappearance normalized to the initial amounts of D1D2 (see the Materials and Methods section). As shown in Fig 2E, the appearance of D1D2 trimers and disappearance of *apo* D1D2 as a function of increasing aVBD molar ratio could be nicely adjusted to linear fits with a Pearson coefficient $R^2 > 0.95$. The CN-PAGE assay was used to determine a rate of D1D2 trimer appearance of 1.51 $AU^{-1}$ ± 0.3 (SEM) and D1D2 monomer disappearance of −1.06 $AU^{-1}$ ± 0.05 (SEM) (Table 1).

### Characterization of D1D2 mutations affecting trimer formation

We estimated that the CN-PAGE assay was sufficiently robust and reproducible to determine potential differences in rates of trimer formation in the various D1D2 variants. All mutations were introduced in D1D2 and the corresponding variants were purified to

**Table 1. Effects of D1D2 mutations on D1D2 complex formation.**

| D1D2 construct | Rate of trimer formation AU$^{-1}$ | Rate of monomer disappearance AU$^{-1}$ | |
|---|---|---|---|
| WT | 1.51 ± 0.03 | −1.06 ± 0.05 | |
| D389R | 0.95 ± 0.06*** | −1.06 ± 0.05$^{n.s.}$ | |
| E143K | 1.10 ± 0.068*** | −0.88 ± 0.10$^{n.s.}$ | Class 1 |
| V367D | 1.19 ± 0.04*** | −0.88 ± 0.11$^{n.s.}$ | |
| K386D | 0.99 ± 0.099*** | −1.29 ± 0.059* | |
| K139E | 1.04 ± 0.08*** | −1.42 ± 0.069* | Class 2 |
| N393D | 1.30 ± 0.05$^{n.s.}$ | −1.32 ± 0.04* | |

A, the rates of trimer formation and monomer disappearance for the indicated D1D2 variants were inferred from linear fits (see the Materials and Methods section). Each value corresponds to the mean of at least three independent experiments ± SD. Class 1: mutations that do not significantly affect the rates of monomer disappearance; class 2: mutations that significantly increase the rates of monomer disappearance. ANCOVA test: *$P < 0.05$; **$P < 0.01$; ***$P < 0.005$. ns, not significant.

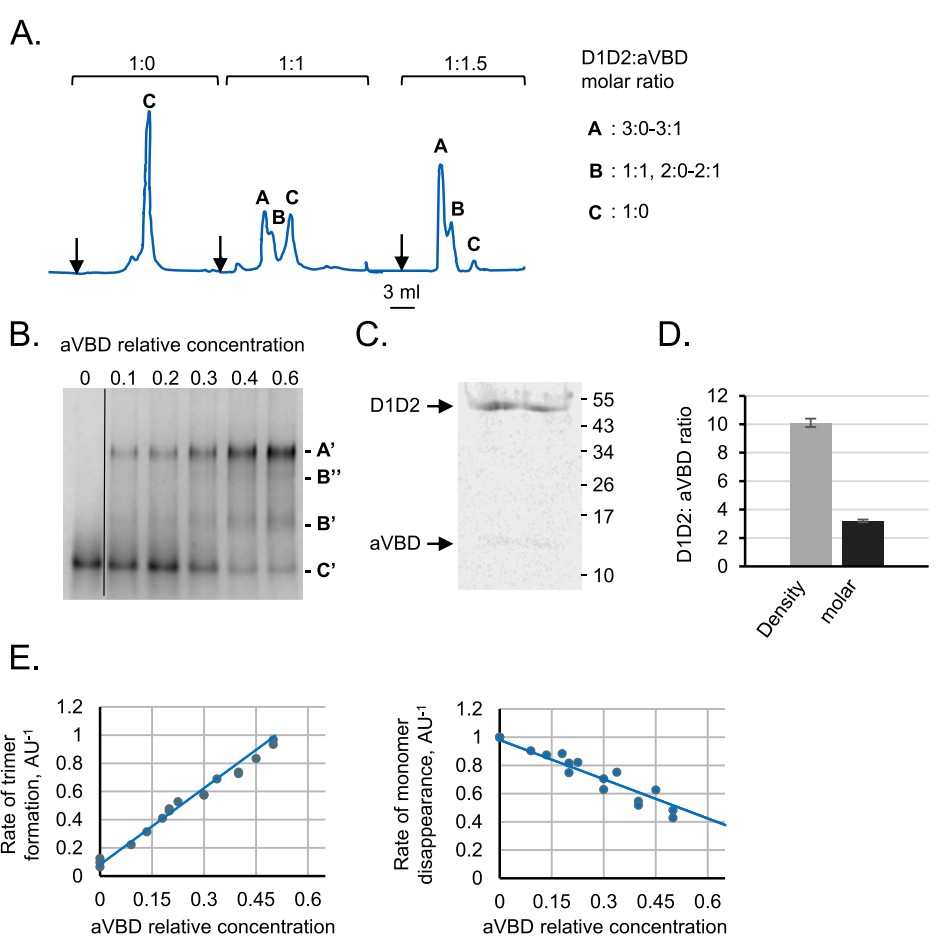

**Figure 2. A CN-PAGE assay to study IpaA-induced D1D2 trimer formation.**
**(A, B)** aVBD and D1D2 were mixed at the indicated molar ratio, with D1D2 at a final concentration of 20 $\mu$M, and incubated for 60 min at 21°C before analysis. **(A)** size exclusion chromatography analysis using an Increase Superdex 200 (see the Materials and Methods section). The indicated stoichiometry is inferred from previous size exclusion chromatography-Multi-Angle Light Scattering analysis (Valencia-Gallardo et al, 2023) and the mass of complexes in peaks A, B, and C estimated from molecular weight standards as a function of the respective elution volume. **(B, D, E)** aVBD relative concentration: ratio of concentrations of aVBD over D1D2. **(B)** CN-PAGE using a 7.5% polyacrylamide native gel followed by Coomassie blue staining. C': monomeric D1D2. A', B' and B'': D1D2 aVBD complexes. The vertical line indicates cutting of the gel to remove irrelevant lanes. **(B, C)** SDS–PAGE followed by Coomassie blue staining of samples corresponding to band A' as shown in Panel (B). **(D)** Densitometry analysis of the bands indicated a D1D2:aVBD ratio in integrated intensity of 10.1 ± 0.36 (SD) corresponding to a 3.2 ± 0.09 (SD) molar ratio consistent with 3:0–3:1 D1D2:aVBD complexes (N = 2). **(E)** the integrated intensity of bands corresponding to trimeric (A') or monomeric D1D2 (C') were scanned by densitometry, and normalized to that of monomeric D1D2 in the absence of aVBD. The lines corresponds to linear fits with a Pearson coefficient > 0.95. The graphs are representative of at least three independent experiments.

homogeneity (Fig S2). Samples were incubated with increasing molar ratio of the aVBD vinculin-binding domain (aVBD) and analyzed by CN-PAGE followed by Coomassie staining (see the Materials and Methods section). As shown in Figs 3 and S3, most of the mutants showed decrease rates of trimer formation. The rates of D1D2 trimer formation and monomer disappearance were inferred from linear fits, following quantification by scanning densitometry (Table 1).

As shown in Table 1, all mutations showed reduced rates of D1D2 trimer and could be distinguished in two classes based on to their effects on the rates of monomer disappearance. Class 1 mutations D389R, E143K and V367D showed no difference in D1D2 monomer disappearance, whereas class 2 mutations K386D and K139E showed increased rates in D1D2 monomer disappearance (Table 1 and Figs 3 and S3). Mutation N393D showed effects similar to the latter mutations, with a slightly reduced rate of trimer formation that

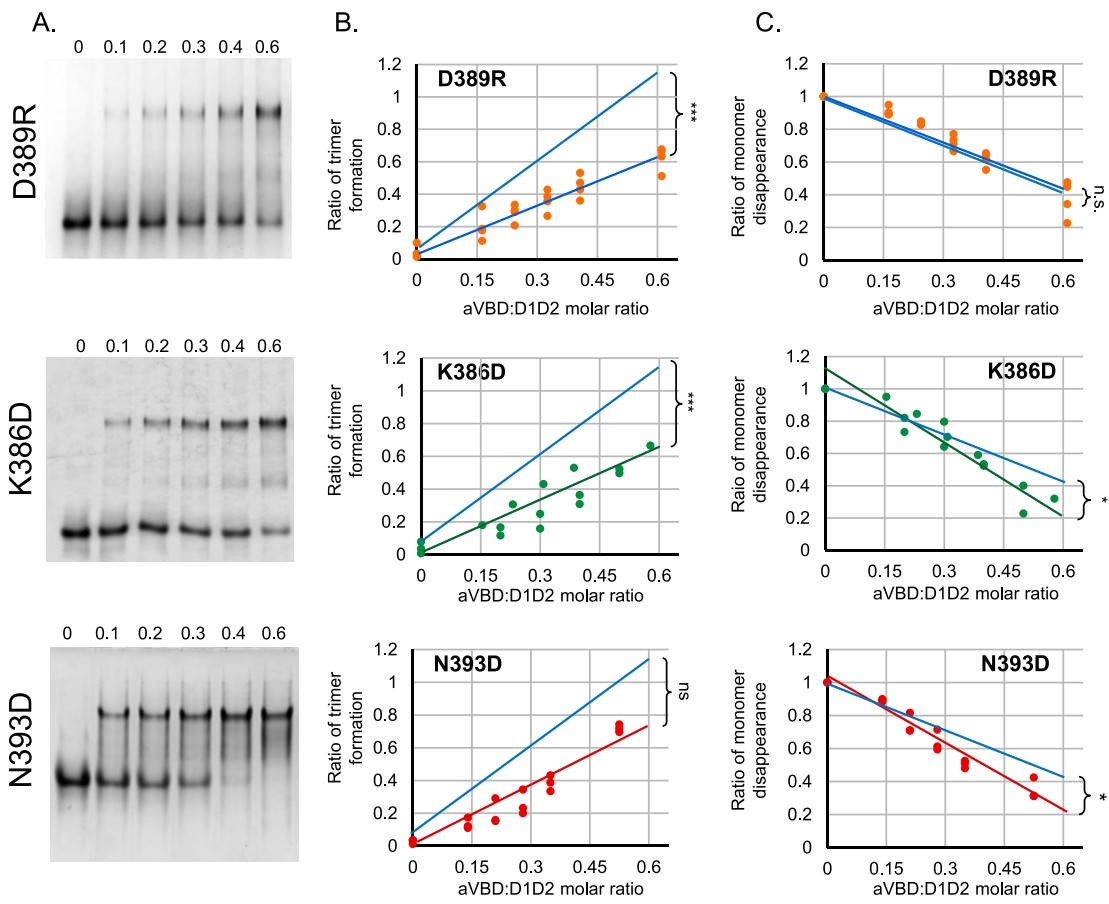

**Figure 3. Effects of mutations on D1D2 trimer formation and monomer disappearance.**
**(A)** CN-PAGE using a 10% polyacrylamide native gel followed by Coomassie blue staining of aVBD-induced complex formation with the indicated D1D2 variant. The aVBD: D1D2 molar ratio is indicated above each lane. **(B, C)** the integrated intensity of bands corresponding to trimeric (B) or monomeric D1D2 (C) were scanned by densitometry, and normalized to that of monomeric D1D2 in the absence of aVBD. The graphs are representative of at least three independent experiments. The lines correspond to linear fits obtained for the sample in the corresponding color. The blue lines correspond to the linear fits obtained for parental D1D2 reported from Fig 2E. ANCOVA test: *$P < 0.05$; ***$P < 0.005$; ns, not significant.

appeared not significant, but increased rates in monomer disappearance (Fig 3A–C). These classes generally correlated with the mutational design based on structural modeling because class 2 and class 1 mutations were expected to target formation of the closed and open D1D2:aVBD 1:1 complex, respectively (Fig 1C and D). The D389R mutation represents an exception, because it was expected to target the D1D2:aVBD 1:1 close complex, but showed no difference in the rate of monomer disappearance (Table 1 and Fig 3A–C).

## D1D2 mutations targeting the close conformer prevent *Shigella* invasion

During *Shigella* invasion, targeting of vinculin by IpaA leads to the cytoskeletal reorganization allowing bacterial attachment to host cells and productive bacterial internalization. In addition, IpaA-mediated vinculin supra-activation triggers the formation of focal adhesions (FAs) distal to the invasion sites that likely contribute to strengthen the adhesion of infected cells even in the absence of mechanotransduction (Valencia-Gallardo et al, 2023). To analyze

the role of D1D2 head-domain oligomerization on *Shigella* invasion, D1D2 mutations were transferred to full-length vinculin fused to mCherry (HV-mCherry) cloned in a eukaryotic expression vector and transfected into MEF vinculin –/– cells (see the Materials and Methods section). In control experiments, Western-blot analysis showed that all constructs were expressed at the expected molecular size at levels comparable with that of WT vinculin-mCherry construct (Fig S4). Transfected cells were challenged with bacteria and bacterial association and internalization were quantified in cells expressing comparable levels of HV-mCherry constructs (see the Materials and Methods section).

As shown in Fig 4, the profiles of bacterial association and invasion were generally similar, consistent with a prominent role in bacterial attachment to the cell during *Shigella* invasion. Consistent for the previously described role for vinculin in *Shigella* invasion, cells complemented with HV-mCherry showed a ca 8-fold and 30-fold increase in bacterial association and invasion compared with mCherry transfected control cells, respectively (Fig 4A–C, HV and mCherry). Similar differences in bacterial association and invasion were observed when comparing with HV-mCherry transfected cells

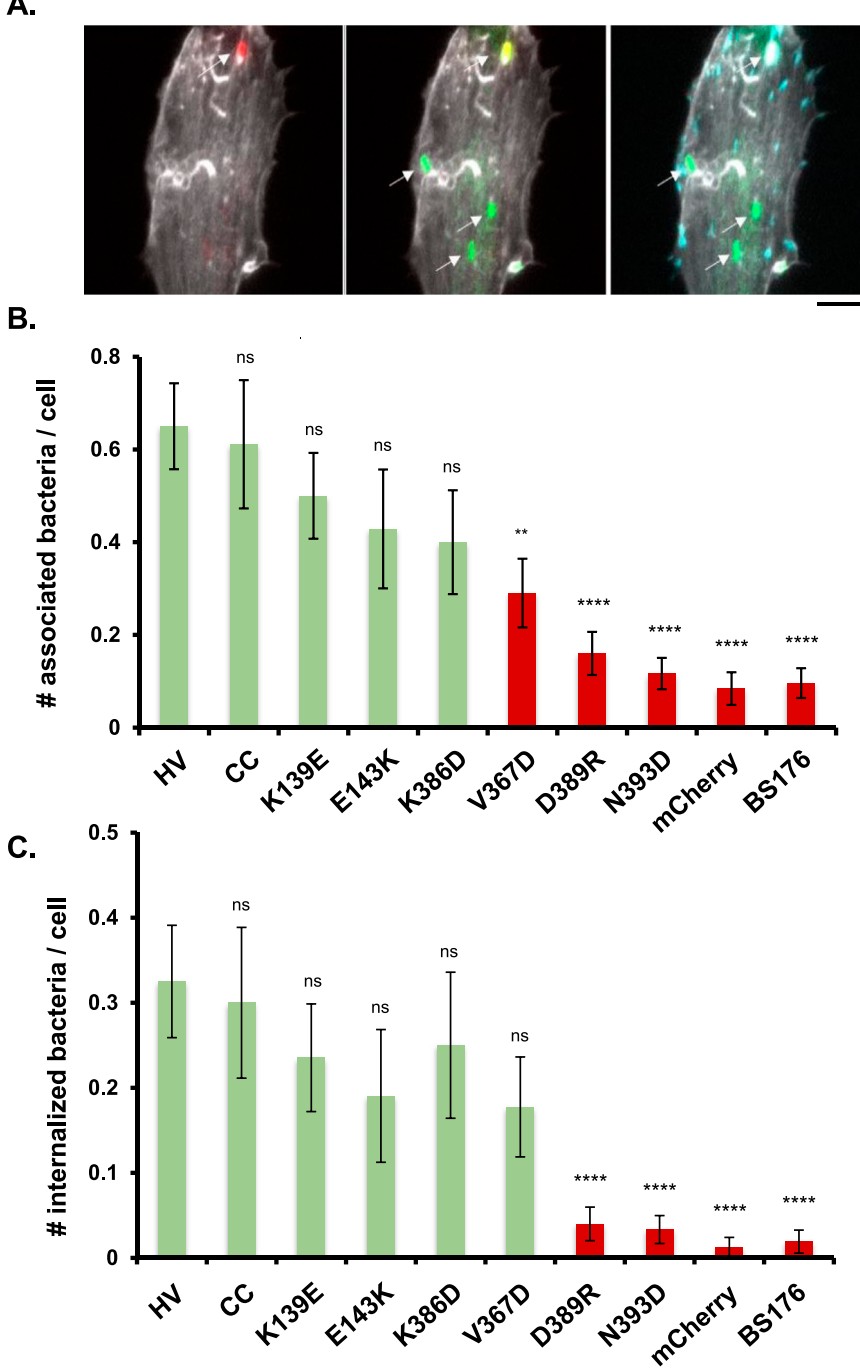

**A.**

LPS

GFP

actin

HV

**B.**

**C.**

**Figure 4. Effects of D1D2 mutations on *Shigella* invasion MEF vinculin −/− cells were transfected with HV-mCherry (HV) or variants bearing the indicated mutations and challenged with GFP-expressing WT *Shigella* for 45 min at 37°C (see the Materials and Methods section).** mCherry: cells transfected with mCherry. BS176: cells transfected with HV and infected with or invasion deficient isogenic derivative strain BS176. After bacterial challenged, samples fixed and processed for bacterial fluorescence inside-out immunostaining and confocal microscopy analysis (see the Materials and Methods section). **(A)** representative maximal projection of fluorescent confocal micrographs of cells transfected with HV and challenged with WT *Shigella*. Staining: green: GFP; red: LPS; cyan: mCherry; gray levels: F-actin. Scale bar: 5 $\mu$m. Note the extracellular bacteria stained in red and green, whereas internalized bacteria stained in green only. **(B)** number of cell-associated bacteria per cell. **(C)** Number of internalized per cell. FA area (n > 30 cells, N = 3). **(C)** average number of FAs per cell (n > 30 cells, N = 3). *t* test: ****$P$ < 0.001; ns, not significant.

challenged with a non-invasive isogenic mutant *Shigella* strain (Fig 4B and C, HV and BS176). Strikingly, mutations D389R and N393D resulted in a drastic reduction in bacterial association and invasion of host cells, to levels comparable with the negative controls (Fig 4B and C). In contrast, the other tested mutations had modest or non-significant effects. These results suggest a key role for polar and charged interactions involving the D389 and N393 at the N-terminal extremity of the H10 helix of D2. The absence of effects associated with the cysteine clamp (CC) and E143K mutations targeting the

open D1D2 conformer suggested that early steps linked to vinculin supra-activation but not later steps linked to vinculin oligomerization are required for *Shigella* invasion.

### Mutations affecting D1D2 trimerization impair focal adhesions' formation

Vinculin oligomerization has been mainly studied in vitro and reported to occur through the Vt-tail domain (Thompson et al, 2013).

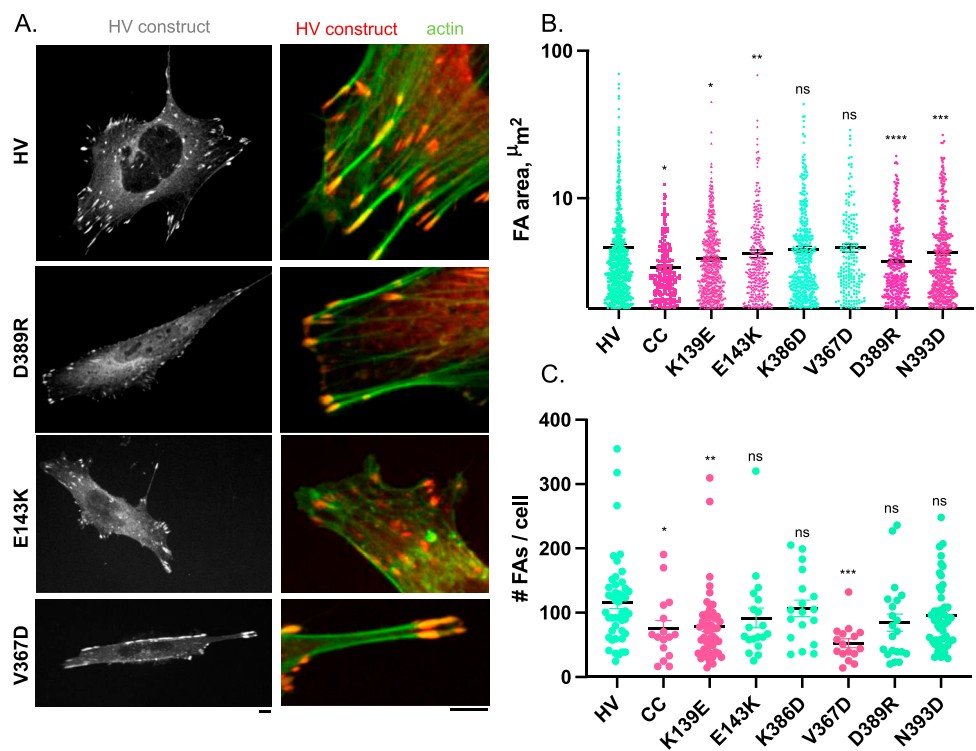

**Figure 5. Effects of D1D2 mutations affecting D1D2 trimer formation on focal adhesions MEF vinculin −/− cells were transfected with HV-mCherry (HV) or variants bearing the indicated mutations.**
Samples were fixed and processed for fluorescence staining of F-actin. **(A)** representative maximal projections of fluorescent confocal micrographs. Green: F-actin; red: HV-mCherry. Scale bar: 5 $\mu$m. **(B)** FA area (n > 1,500, N = 2). **(C)** average number of FAs per cell (n > 20, N = 2). Mann-Whitney: *$P$ < 0.05; **$P$ < 0.01; ***$P$ < 0.005. ns, not significant.

In cells, the role of vinculin oligomerization remains unclear. We therefore tested the effects of D1D2 mutations of FA formation, a process relevant for bacterial infection of host cells because it controls the adhesion/detachment of *Shigella*-infected cells from the extracellular matrix.

As shown in Fig 5, all tested mutations induced a decrease either in FA area or FA number per cell (frequency) or both, although to variable extent (Figs 5 and S5A–C). The K386D mutation showed a decrease in FA area and frequency but the difference relative to parental HV was not statistically significative, as opposed to the other mutations (Fig 5B and C). As previously reported, the CC mutation altered the size and frequency of FAs (Fig 5; Valencia-Gallardo et al, 2023). The K139E mutation also significantly affected both FA size and frequency. The E143K, D389R and N393D mutations affected the size of FAs, whereas the difference in FA number per cell relative to parental HV was not significant because of the large dispersion of values (Fig 5B and C). Interestingly, the V367D variant also did not show a significant difference in the average FA size compared with control cells expressing parental mCherry-vinculin (Fig 5B and C), despite a significant reduction in the number of FAs per cell. The FAs in the V367D variant, although similar in size, looked different than those in control cells, being mostly at the cell periphery and particularly elongated (Fig 5A). This qualitative difference was confirmed by quantification of the AR index, showing a pronounced difference for the V367D variant compared with the other samples (Fig S5B and C).

Together, these results support a role for vinculin supra-activation and head-domain–mediated oligomerization in the formation of focal adhesions independent of *Shigella* invasion.

## IpaA VBSs prevents cell motility

Our findings suggested that during *Shigella* invasion, IpaA-mediated vinculin head-domain oligomerization played mostly a role in up-regulating cell adhesion rather than bacterial invasion. Vinculin, however, is paradoxically described as a prognostic marker favoring the migration of cancer cells or as a tumor suppressor stimulating cell anchorage (Mierke et al, 2010; Hamidi & Ivaska, 2018). These contradictory findings reflect its complex and poorly understood regulation, as well as different roles in 2D or 3D systems (Gulvady et al, 2018).

We therefore tested the effects of IpaA on the motility and invasion of melanoma cells. In time-lapse microscopy experiments in 2D-chambers, GFP-aVBS1-2 inhibited melanocyte motility compared with control cells, with a rate of Root Median Square Displacement (rMSD) of 3.16 and 15.6 $\mu$m.min$^{-1}$, respectively (Fig S6A and B). An even stronger inhibition was observed for GFP-aVBD transfected cells (rMSD = 2.3 $\mu$m.min$^{-1}$) (Fig S6A and B). Transmigration of melanocytes in 3D-matrigels was similarly inhibited by aVBS1-2 and aVBD (Fig S6C).

These results suggest that IpaA-mediated vinculin "canonical" activation induced by aVBS1-2, as well as "supra-activation" mediate by aVBD do not stimulate cell motility but rather strengthen cell adhesion, consistent with their described effects of FAs (Valencia-Gallardo et al, 2023).

## Vinculin supra-activation catalyzed by IpaA strengthens cell adhesion

We next investigated the role of vinculin supra-activation by comparing the effects of aVBS1-2 and aVBD in dynamic cell

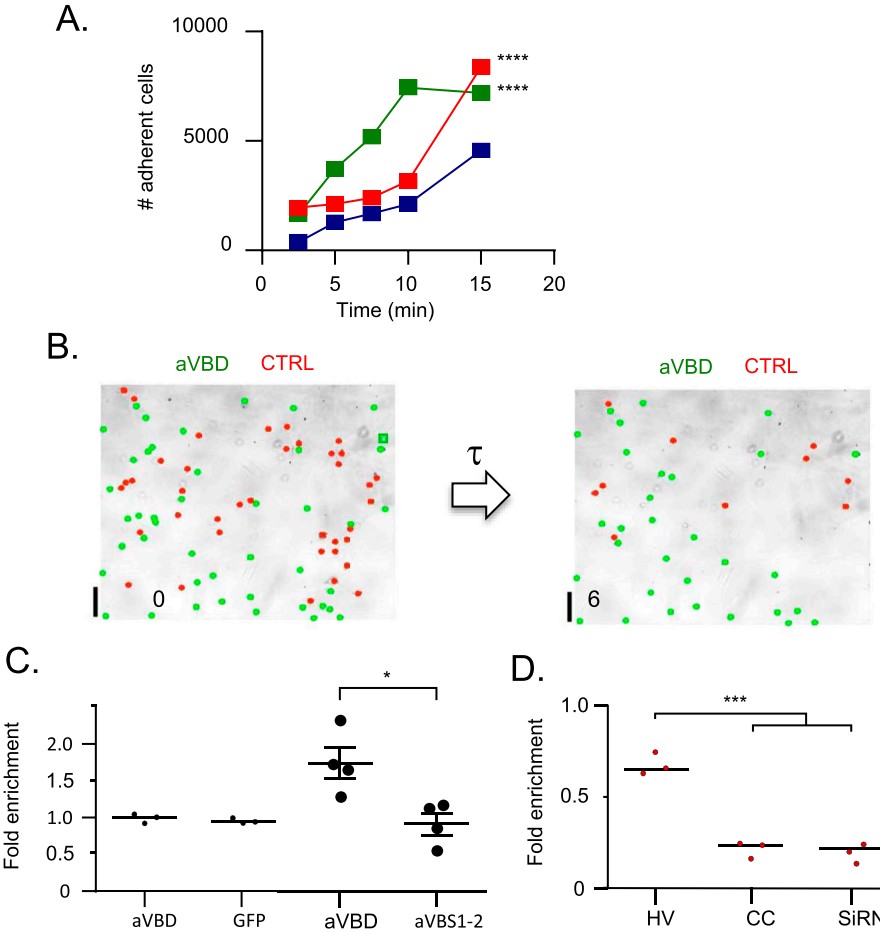

**Figure 6. aVBD-mediated vinculin supra-activation stimulates rapid cell adhesion.**
**(A)** 1205Lu melanoma cells were transfected with GFP alone (blue), GFP-aVBS1-2 (red) or GFP-aVBD (green), lifted up by trypsinization and plated for the indicated time on Fn-coated coverslips. Samples were washed, fixed and adherent cells were scored microscopically. The total number of adherent cells scored is indicated. GFP: 3,223 cells, N = 4; GFP-aVBS1-2: n = 7,418, N = 4; GFP-aVBD: n = 5,668, N = 4. Chi square corrected with Bonferroni mutliple comparison correction. ****$P$ < 0.05. **(B, C, D)** 1205Lu melanoma cells were transfected with the indicated constructs, labeled with calcein (see the Materials and Methods section) and mixed with the same ratio of control cells. **(C, D)** Cells were perfused in a microfluidic chamber and allowed to adhere for: (C), 30 min (small solid circles) and 15 min (large solid circles) before shear stress application; or (D), 20 min. **(B)** representative fields. The number indicates the elapsed time in minutes after shear stress application. **(C, D)** Scatterplot of fold enrichment of: aVBD (N = 4, n = 610) or aVBS1-2 (N = 4, n = 433) transfected cells versus control cells (1,594 cells, N = 4). **(D)** aVBD (N = 3, n = 557); GFP (N = 3, n = 490); HV: vinculin mCherry (N = 3, n = 481); CC: vinculin Q68C A396C-mCherry (N = 3, n = 259); siRNA: cells treated with anti-vinculin siRNA (N = 3, n = 395). Unpaired $t$ test. ***$P$ < 0.005.

adhesion experiments. First, we performed cell replating experiments, where resuspended transfected cells were allowed to attach to fibronectin-coated surfaces for controlled time periods. As shown in Fig 6A, aVBD induced higher yields of adherent cells when replating was performed with short kinetics, with a fivefold increase over control cells or cells transfected with aVBS1-2 after 10 min replating. By contrast, little difference in adhesion yield was detected between samples after 15 min replating suggesting that aVBD predominantly affected the early dynamics rather than increasing cell adhesion (Fig 6A). To extend these findings, we measured cell adhesion strength using controlled shear stress in a microfluidic chamber (Fig 6B and Video 1). Consistent with replating experiments, when cells were allowed to adhere to fibronectin-coated surfaces for 30 min, little difference in resistance to shear stress could be detected between GFP-aVBD and GFP-transfected cells samples (Fig 6C, small solid circles). However, when shear stress was applied 15 min after cell incubation, GFP-aVBD-transfected cells showed significantly higher resistance to shear stress up to 22.2 dyn.cm$^{-2}$ compared with GFP-aVBS1-2 or GFP-transfected cells, with 1.7 ± 0.2- and 0.9 ± 0.14-fold enrichment ± SD of adherent cells for GFP-aVBD and GFP-aVBS1-2 transfected cells, respectively (Fig 6C, large solid circles, Video 1). Consistent with a role for vinculin oligomerization in adhesion, cells transfected with the cysteine-clamped vinculin variant showed adhesion levels under shear

stress comparable with cells depleted for vinculin by siRNA treatment that were decreased threefold compared with WT vinculin-transfected cells (Figs 6D and S7). These results are in full agreement with effects observed on adhesion structures and suggest that aVBD-mediated vinculin supra-activation accelerates endogenous processes occurring during mechanotransduction to promote strong adhesion.

# Discussion

In this work, we used structural models of conformers based on mass spectrometry analysis of cross-linked 1:1 D1D2:aVBD complexes to identify polar residues of the H10 helix of D2 involved in D1D2 trimerization. We found that the effects on the in vitro trimerization of purified D1D2 variants observed in native-gel shift assays to be generally in line with the predicted conformer structures, with all mutations affecting IpaA-induced D1D2 trimer formation.

We found that mutations either had no effects (class 1 mutations) or increased the rates of D1D2 monomer disappearance (class 2 mutations). None of the mutations delayed D1D2 monomer disappearance, indicating that they did not prevent the formation

of a 1:1 D2D2:aVBD complex. These results are expected because IpaA VBS1-2 drives the initial binding to D1D2, independent of IpaA VBS3 (Tran Van Nhieu & Izard, 2007). Thus, the mutations designed to interfere with IpaA VBS3 binding to D1D2, likely affect allosteric changes leading to D1D2 oligomerization without affecting the formation of a 1:1 D1D2:aVBD complex. There are different explanations for the effects of K386D and N393D class 2 mutations that increase the rates of D1D2 monomer disappearance. These mutations may increase the rates of IpaA binding to D1D2 or may stabilize an allosteric D1D2 conformer subsequent to the formation of a 1:1 D1D2:aVBD 1:1 complex. The effects of the K139E mutation that also accelerates monomer disappearance may be explained different manner. Indeed, in apoD1D2, K139 establish a salt bridge with D389 that likely stabilize the D1D2 in the close conformation. The vinculin K139E charge inversion may therefore destabilize D1D2 to favor the formation of intermediate D1D2:aVBD complexes.

We found that the V367D predicted to disrupt the putative coiled-coil heptad motif in D2 H10-affected trimer formation, suggesting a role for this motif in IpaA-induced D1D2 oligomerization. In the structures of apo D1D2 of full-length vinculin, this motif is located at a face of H10 buried into the D2 second helix bundle both in the apo/closed and open D1D2 conformer. Exposure of this putative coiled-coil motif would require major unraveling of the D2 bundles that was not observed when using atomic force microscopy and steered molecular dynamics with pulling forces ranging from 30 to 60 pN at a single vinculin molecule level (Kluger et al, 2020). It is possible, however, that higher forces expected from bundling of actin filaments pulling on D1D2 trimers could result in D2 unfolding, leading to exposure of the coiled-coil motif and vinculin higher order oligomerization.

In Fig S8, we depicted potential steps of aVBD-induced D1D2 trimerization affected by the mutations. We posited that class 2 mutations stimulated the transition to the closed 1:1 D1D2:aVBD complex immediately occurring upon aVBD binding to apo D1D2, whereas inhibiting the subsequent step leading to the open D1D2 conformer (Fig S8). In contrast, the class 1 E143E and cysteine-clamp mutations that did not affect the rates of monomer disappearance may target a later step associated with the "opening" of the D1D2 subdomains. The V367D "coiled-coil" mutation could also target a later step impairing D1D2 oligomerization (Fig S8). It is possible that class 2 mutations favor the formation of D1D2 dimers that do not trimerize, such as those induced by aVBS1-2 (Valencia-Gallardo et al, 2023). The D389R mutation may mimic the effects of IpaA K499 on the closed D1D2 conformer, therefore explaining the absence of difference on aVBD-induced monomer disappearance, whereas preventing transition to the open D1D2 conformer (Fig S8).

Our previous findings based on a clamped-mutant of vinculin proficient for canonical activation but deficient for supra-activation suggest that the head-domain–mediated oligomerization of vinculin also contributes to the maturation of focal adhesions (Valencia-Gallardo et al, 2023). Consistent with a role for vinculin-mediated head domain oligomerization, when transferred to full-length vinculin, we found that these mutations affecting D1D2 trimerization also affected the formation of focal adhesions. We did not find, however, a clear correlation between the class 1 and class 2 mutations expected to affect different steps of oligomerization in vitro and distinct defects on focal adhesion formation. Our

analysis, however, was limited to the size or number of focal adhesions per cell. A more detailed analysis including measurements of traction forces exerted by focal adhesions could enable a better characterization of these vinculin mutations affecting head domain trimerization. Also, in cells vinculin activation is regulated by various process including phosphorylation and binding to PI 4,5 $P_2$ that may also impact on its oligomerization.

Of note, the size reduction in focal adhesions associated with D1D2 mutations was more specifically observed at the levels of peripheral adhesions, whereas little difference was observed for ventral adhesions relative to parental vinculin. Ventral adhesions appeared thin and elongated, consistent with fibrillar adhesions known to remain stable independent of force (Sun et al, 2016). These observations are consistent with head domain-mediated vinculin oligomerization occurring at high force regime. As opposed to other mutations, the V367D mutation led to a drastic reduction in the number of focal adhesions but did not affect their average size. Instead, the V367D variant mostly formed elongated adhesions at the cell periphery. Whereas the role of the putative coiled-coil domain targeted by the V367D mutation deserves clarification, the effects on the focal adhesion morphology associated with this particular mutation is consistent with the impairment in a process during later steps of vinculin oligomerization and focal adhesion formation that is different than the other mutations (Fig S8).

We found that as opposed to the effects observed for all D1D2 mutations on focal adhesions, only the N393D and D389R mutations affected *Shigella* invasion of host cells. These mutations are predicted to impair the interaction of IpaAVBS3 in the closed but not in the open D1D2:aVBD conformer. Because in this closed conformer, D1D2 adopts a conformation similar to that of D1D2:aVBS1-2 or apo D1D2, this suggests that it occurs before the formation of the open D1D2:aVBD conformer that leads to vinculin head-domain oligomerization (Fig 4B; Valencia-Gallardo et al, 2023). In contrast, D1D2 mutations targeting the open conformer and vinculin oligomerization had no significant effects on bacterial invasion. The results suggest that IpaA-mediated vinculin supra-activation and head-domain oligomerization are not critical for *Shigella* invasion but are required to strengthen matrix adhesion of bacteria-infected cells. Accordingly, rather than acting at the bacteria-cell contact sites where it is injected by the *Shigella* T3SS, IpaA would induce the formation of vinculin oligomers diffusing at a distance from the supra-activation site to reinforce cell adhesions at basal membranes. This view is in line with the "catalytic" action of IpaA on vinculin oligomer formation observed in vitro (Valencia-Gallardo et al, 2023).

## Materials and Methods

### Bacterial strains, cells, and plasmids

The bacterial strain used for the purification of the D1D2 construct is *E. coli* BL21 (DE3) from Invitrogen. *E. coli* DH5-α F− *endA1 glnV44 thi-1 recA1 relA1 gyrA96 deoR nupG purB20 φ80dlacZΔM15 Δ(lacZYA-argF) U169, hsdR17(rK−mK+), λ−* was used for the purification of aVBD. The

**Table 2. Primers used in this study.**

| D1D2 mutation | Primers |
|---|---|
| D389R | 5'-CATTGCAAAGAAGATCCGTGCTGCTCAGAACTGGC-3' |
| | 5'- GCCAGTTCTGAGCAGCACGGATCTTCTTTGCAATG-3' |
| E143K | 5'- AGGAATTTTGAAATATCTTACAGTG-3' |
| | 5'- CACTGTAAGATATTTCAAAATTCCT-3' |
| K386D | 5'- GAGCATTGCAGACAAGATCGATGC-3' |
| | 5'- GCATCGATCTTGTCTGCAATGCTC-3' |
| K386G | 5'- CAAAGCAGAGCATTGCAGGCAAGATCGATGCTGCTC-3' |
| | 5'- GAGCAGCATCGATCTTGCCTGCAATGCTCTGCTTTG-3' |
| K139E | 5'- TAGAGTTTGCGAAGGAATTTTGG-3' |
| | 5'-CCAAAATTCCTTCGCAAACTCTA-3' |
| V367D | 5'-CTCACAGCAAAAGTGGAAAATGCAGCTCGC-3' |
| | 5'-GCGAGCTGCATTTTCCACTTTTGCTGTGAG-3' |
| N393D | 5'-GATCGATGCTGCTCAGGACTGGCTTGCAGATCCAAATG-3' |
| | 5'-CATTTGGATCTGCAAGCCAGTCCTGAGCAGCATCGATC-3' |
| N379E | 5'- GCTGGAAGCCATGACCGAATCAAAGCAGAGCATTGC-3' |
| | 5'- GCAATGCTCTGCTTTGATTCGGTCATGGCTTCCAGC-3' |

GFP-expressing WT serotype V *Shigella* strain M90T and isogenic plasmid-cured non-invasive derivative BS176 were described previously (Valencia-Gallardo et al, 2019).

MEF and MEF vinculin null cells (Humphries et al, 2007) were grown in DMEM 1g/liter glucose containing 10% FCS in a 37°C incubator containing 10% $CO_2$.

The pGEX-4T2-aVBD and the pET15b-D1D2 plasmids were described previously. The mutations in D1D2 were introduced by site-directed mutagenesis using pET15b-D1D2 as a matrix and the primer pairs indicated in Table 2. The pmCherry-N1-human vinculin (HV-mCherry) and was from Addgene. Mutations in D1D2 were transferred in mCherry-HV by exchanging the NheI-PspXI fragment with the corresponding XbaI-PspXI fragment of pET15b-D1D2. The CC mutation corresponds to substitutions of glutamine 69 and Alanine 386 of human vinculin by cysteine residues preventing the formation of IpaA-induced vinculin oligomers but not F-actin binding upon vinculin activation (Valencia-Gallardo et al, 2023).

MEF vinculin null cells (Humphries et al, 2007) were routinely grown in DMEM 1g/liter glucose containing 10% FCS in a 37°C incubator containing 10% $CO_2$.

The 1205Lu melanoma cell line, a gift from Dr. M. Herlyn (Wistar Institute, Philadelphia, PA), was cultured in RPMI supplemented with 10% FCS as described (Javelaud et al, 2005).

## Protein purification

BL21 (DE3) competent *E. coli* was transformed with the pET15b-D1D2 variant constructs. D1D2 were purified as described (Park et al, 2011b). For the IpaA derivatives, DH5-a competent *E. coli* was transformed with pGEX-4T2-aVBD. Bacteria were grown at 37°C with shaking until $OD_{600nm}$ = 1.0 were induced with 1 mM IPTG and incubated for another 2 h. Bacteria were pelleted and washed in ice-cold lysis buffer containing 25 mM Tris PH 7.4, 100 mM NaCl and 1 mM beta-mercaptoethanol, containing complete protease inhibitor. All subsequent steps were performed at 4°C. Bacterial pellets were resuspended in 1/20th of the original culture volume and lyzed using a high pressure homogeneizer (LM20, Microfluidics Corp.). Cell debris were pelleted by centrifugation at 8,000 g for 20 min. Clarified lysates were subjected to affinity chromatography using a GSTrap HP affinity column (GE Healthcare). Briefly, after incubation with the clarified lysates, the column was washed with five column volumes before incubation in PBS containing 100 µg/ml Thrombin (ref 27084601; Cytiva) for 16 h at 21°C. aVBD was then eluted in PBS and further subjected to purification using SEC using a Superdex 200 10/300 GL (Ge Healthcare). Samples were stored aliquoted at –80°C at concentrations ranging from 1 to 10 mg/ml.

## SEC analysis

D1D2 and aVBD were at the indicated molar ratio, with D1D2 at a final concentration of 20 µM, for 60 min at 21°C. 200 µl of the protein mixtures were analyzed by SEC on a Superdex 200 10/300 GL (GE Healthcare) using a GE ÄKTA FPLC (Fast Protein Liquid Chromatograph, GMI) and a collection volume of 200 µl per fraction and 20 ml of total collected volume. The SEC buffer was 25 mM Tris–HCl pH 7.2, 100 mM NaCl.

## CN-PAGE and densitometry analysis

The D1D2 variants and aVBD were at the indicated molar ratio, with D1D2 at a final concentration of 20 µM, for 60 min at 21°C. Protein complex formation was visualized by PAGE under non-denaturing conditions using à 7.5% polycrylamide gel, followed by Coomassie blue staining, as described previously (Wittig & Schägger, 2005).

Gel raw images were aquired with a ChemiDoc Imaging System (Bio-Rad Laboratories, Inc.) using the Coomassie staining settings. Band intensities corresponding to the monomer, higher order oligomer or intermediate species were measured with Fiji-ImageJ using a fixed rectangular area adjusted with the max width for the monomer and max high order oligomer for the height (Schindelin et al, 2012). A reference area outside the lanes used for the running reactions was included to subtract the background. Values were plotted and then normalized to the value obtained for the corresponding apo construct.

## Cell transfection

For transfection experiments, cells were seeded at $1 \times 10^4$ cells on 25 mm-diameter coverslips coated with fibronectin at a concentration of 20 µg/ml. Cells were transfected with 1 µg of the pGEX-4T2-D1D2 construct and 4 µls JetPEI transfection reagent (Polyplus) for 16 h following the manufacturer's recommendations.

## Bacterial association and invasion assays

GFP-expressing bacteria were inoculated from pre-cultures at a dilution of 1:100 in trypticase soy broth containing chloramphenicol at 15 µg/ml and grown in a shaking incubator at 37°C until reaching an optical density at 600 nm ($OD_{600\ nm}$) of 0.4. Bacterial cultures

were centrifuged at 7,500g for 10 min and the bacterial pellet was resuspended in EM buffer (120 mM NaCl, 7 mM KCl, 1.8 mM $CaCl_2$, 0.8 mM $MgCl_2$, 5 mM glucose, and 25 mM Hepes, pH = 7.3) at a final $OD_{600\ nm}$ of 0.1. Cells were incubated with the bacterial suspension for 60 min at 37°C before fixing and processing for fluorescence microscopy analysis. Samples were fixed in PBS containing 3.7% paraformaldehyde for 645 min at 21°C, before processing for fluorescence staining of F-actin using Phalloidin-Alexa 488. Extracellular and internalized bacteria were scored using an "inside-out" staining procedure as previously described (Valencia-Gallardo et al, 2023). Briefly, samples were incubated for 45 min at RT with a rabbit anti-LPS antibody diluted 1:5,000 in PBS in the absence of permeabilization, followed by incubation with an anti-rabbit IgG antibody coupled to Alexa Fluor 635 diluted 1:200 to label extracellular bacteria. Samples were then permeabilized by incubation with PBS containing 0.1% Triton X-100 for 4 min, before incubation with Phalloidin-Alexa 350 diluted 1:5,000 to label F-actin. Samples were mounted in DAKO mounting media. Bacterial association with cells was determined by scoring all GFP-expressing bacteria. Bacterial invasion was determined by scoring GFP-expressing bacteria that did not label with Alexa Fluor 650. Bacterial counts were normalized to the cell number.

### Fluorescence confocal microscopy analysis

For *Shigella* invasion assays, samples were analyzed using a Leica DMIRBe microscope equipped with a 63/1.25 HCX PL APO immersion objective, and a CoolSnap HQ2 camera controlled by the Metamorph software 7.7 from Roper Scientific Instruments. For each field, 24–30 Z-planes spaced by 400 nm were acquired in stream mode for each wavelength. The bacterial association rates were determined by scoring GFP-expressing bacteria in contact with cells. The bacterial invasion rates were determined by scoring GFP-expressing bacteria that were not labeled with the anti-LPS antibody. For focal adhesion, samples were analyzed using an Eclipse Ti inverted microscope (Nikon) equipped with a 60 x objective APO TIRF oil immersion (NA: 1.49), a CSU-X1 spinning disk confocal head (Yokogawa), and a Prime 95B CMOS camera (Photometrics) controlled by the Metamorph 7.7 software.

### Image analysis

Focal adhesions were analyzed using the ImageJ 2.1.0/1.53c software. Cells expressing similar levels of average fluorescence intensity of the HV-mCherry constructs were analyzed. For each set of experiments, the confocal plane corresponding to the basal plane was subjected to thresholding using strictly identical parameters between samples. Adhesion clusters were detected using the "Analyze particle" plug-in, setting a minimal size of 3.5 $\mu m^2$.

### Statistical analysis

The number of adhesions was analyzed using Dunn's multiple comparisons test. The median area was compared using Mann-Whitney test. Differences in the rates of D1D2 trimer formation and monomer disappearance based were analyzed using an ANCOVA test.

### Live cell tracking

1205Lu melanoma cells were transfected with IpaA constructs or GFP alone (control) and transferred in microscopy chamber on a 37°C 5%-$CO_2$ stage in RPMI1640 medium, containing 25 mM Hepes. For cell tracking, samples were analyzed using and inverted Leica DMIRBe microscope and a 20 X phase contrast objective. Image acquisitions were performed every 3 min for 200 h. The mean velocity of migration was measured for all tracks followed for at least 5 h. The root square of MdSD over time was plotted over time and fitted by linear regression. The slopes of the linear fit were compared using an ANCOVA test (linear model). The median cell surface was quantified as the mean of the surface for three time points (25%, 50%, and 75%) of the whole cell track and dispersion measured by the median absolute dispersion.

### Cell invasion assays

Tissue culture Transwell inserts (8 $\mu m$ pore size; Falcon) were coated for 3 h with 10 $\mu g$ of Matrigel following the manufacturer's instructions (Biocoat, BD Biosciences). Inserts were placed into 24-well dishes containing 500 $\mu l$ of RPMI medium supplemented with 1% FCS. $5 \times 10^4$ melanoma cells were added to the upper chamber in 250 $\mu l$s of serum-free RPMI medium. After 24 h, transmigrated cells were scored by bright field microscopy. Experiments were performed at least three times, each with duplicate samples.

### Microfluidics cell adhesion assay

Analysis of cell detachment under shear stress was based on previous works (Ghannam et al, 2011). 1205Lu melanocytes were transfected with the indicated constructs, then labeled with 2 $\mu M$ calcein-AM (Life Technologies) in serum-free DMEM for 20 min. Cells were detached by incubation with 2 $\mu M$ Cytochalasin D (Sigma-Aldrich) for 40 min to disassemble FAs, followed by incubation in PBS containing 10 mM EDTA for 20 min. Cells were washed in EM buffer (120 mM NaCl, 7 mM KCl, 1.8 mM $CaCl_2$, 0.8 mM $MgCl_2$, 5 mM glucose and 25 mM Hepes at pH 7.3) by centrifugation and resuspended in the same buffer at a density of $1.5 \times 10^6$ cells/ml. Calcein-labeled transfected cells and control unlabeled cells were mixed at a 1:1 ratio and perfused onto a 25 mm-diameter glass coverslips (Marienfeld) previously coated with 20 $\mu g$/ml fibronectin and blocked with PBS containing 2% BSA (Sigma-Aldrich) in a microfluidic chamber on a microscope stage at 37°C. We used a commercial microfluidic setup (Flow chamber system 1C, Provitro) and a Miniplus3 peristaltic pump (Gilson) to adjust the flow rate in the chamber. Microscopy analysis was performed using a LEICA DMRIBe inverted microscope equipped with a Cascade 512B camera and LED source lights (Roper Instruments), driven by the Metamorph 7.7 software (Universal imaging). Cells were allowed to settle for the indicated time before application of a 4 ml/min, flow corresponding to a wall shear stress of 22.2 $dyn/cm^2$ (2.22 Pa). Acquisition was performed using a 20 X objective using phase contrast and fluorescence illumination (excitation 480 ± 20 nm, emission 527 ± 30 nm). Fluorescent images were acquired before and after flushing to differentiate between target and control cells. Phase contrast images were acquired every 200 ms. Fold

enrichment was defined as the ratio between of attached labeled and unlabeled cells.

## Supplementary Information

## Acknowledgements

The research was supported by fundings from the Inserm and CNRS. G Tran Van Nhieu is a recipient of the grant ANR-21-CE35-0007-03 CALPLYCX. DI Aguilar and B Cocom-Chan received funding from the CONACYT. H Khakzad was supported by the French Agence Nationale de la Recherche (ANR), under grant ANR-22-CPJ2-0075-01.

### Author Contributions

B Cocom-Chan: data curation, investigation, and methodology.
H Khakzad: conceptualization, data curation, and methodology.
M Konate: methodology.
DI Aguilar: methodology.
C Bello: methodology.
C Valencia-Gallardo: data curation and methodology.
Y Zarrouk: methodology.
J Fattaccioli: formal analysis and writing—review and editing.
A Mauviel: formal analysis and writing—original draft.
D Javelaud: methodology and writing—review and editing.
G Tran Van Nhieu: conceptualization, data curation, methodology, and writing—original draft, review, and editing.

### Conflict of Interest Statement

The IpaA constructs in this study are associated with the patent NoPVT/EP2016/073287.2016 with no restriction for academic research but with restriction for commercial use.

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
