## [Reviewer comments · Life Science Alliance]

IpaA reveals distinct modes of vinculin activation during *Shigella* invasion and cell-matrix adhesion

Benjamin COCOM CHAN, Hamed KHAKZAD, Mahamadou KONATE, Daniel Isui Aguilar, Chakir BELLO, Cesar Margarito VALENCIA-GALLARDO, Yosra ZARROUK, Jacques Fattaccioli, Alain Mauviel, Delphine Javelaud and Guy Tran Van Nhieu

DOI: <https://doi.org/10.26508/lsa.202302418>

Corresponding author(s): Dr. Guy Tran Van Nhieu (Institute of Integrative Biology of the Cell)

Review Timeline:

Submission Date:	2023-10-05
Editorial Decision:	2023-12-13
Revision Received:	2024-04-29
Editorial Decision:	2024-05-20
Revision Received:	2024-05-23
Accepted:	2024-05-24

Transaction Report:

December 13, 2023

Re: Life Science Alliance manuscript #LSA-2023-02418-T

Dr. Guy TRAN VAN NHIEU
I2BC UMR9198 U1280
Cell Biology
1 avenue de la Terrasse
ESSONNE 91190
France

Dear Dr. TRAN VAN NHIEU,

Thank you for submitting your manuscript entitled "IpaA reveals distinct modes of vinculin activation during Shigella invasion and cell-matrix adhesion". The manuscript has been evaluated by expert reviewers, whose reports are appended below. Unfortunately, after an assessment of the reviewer feedback, our editorial decision is against publication in Life Science Alliance.

Although your manuscript is intriguing, I feel that the points raised by the reviewers are more substantial than can be addressed in a typical revision period. If you wish to expedite publication of the current data, it may be best to pursue publication at another journal.

Given the interest in the topic, I would be open to re-submission to Life Science Alliance of a significantly revised and extended manuscript that fully addresses the reviewers' concerns and is subject to further peer review. If you would like to resubmit this work to Life Science Alliance, you may submit an appeal directly through our manuscript submission system. Please note that priority and novelty would be reassessed at re-submission.

Regardless of how you choose to proceed, we hope that the comments below will prove constructive as your work progresses.

Thank you for thinking of Life Science Alliance as an appropriate place to publish your work.

Sincerely,

Reviewer #1 (Comments to the Authors (Required)):

Cocom-Chan et. al. provide an analysis of the Shigella IpaA-vinculin interaction. The profusive use of abbreviations significantly impairs the readability and an ability to interpret the authors conclusions. The figure references are wrong throughout and made this reviewer have to guess whether the statements presented in the text were associated with particular figures. On its own the inability to read the manuscript and connect the figures to statements significantly limits the enthusiasm for the study.

1. Fig. 1A - left: shows two D2 domains and no D3 Domain. If this is correct, could you clarify what is being depicted in the legend?
2. Fig. 1A - right: it is difficult to interpret what is happening. Are there coils drawn onto the structure for IpaA? It appears there is something there in green above the arrow for VBS2, but it is very difficult for me to see it clearly. I would either make these clearer as in panel B or remove them. Are the colors that are present on the left panel the same as in the right or is one of the 3D constructions supposed to represent IpaA?
3. The results state " Structural models indicate that aVBD triggers a 30 % angle displacement of the major axis of the D1 and D2 subdomains relative to the apo D1D2 or D1D2 in complex with IpaA VBS1-2 only (Fig. 1B;...", Yet Fig. 1B shows complex with all three VBS domains. Please clarify what is happening in the figure. What do the authors mean by aVBD in this legend. There is no aVBD on the figure. And it is not defined in the legend.
4. Why does the arrangement of the vinculin domains in panel A seem to not match the arrangement of the domains in panel B when the 3 domains are interacting. Please clarify what it happening. It seems to suggest that the D1-D2 interaction studies to not match the previous crystal structure of all of the domains? It appears the interactions in panel A match the domain orientation when there is only VBS1 and 2 binding, but all three VBS interactions are presented?
5. "As shown in Fig. 1B, in the closed D1D2:aVBD complex, IpaA VBS3 mainly interacts with the H10 helix of D2." No H10 helix

- is in Fig. 1B. Moreover, there is no "H10 helix" label anywhere in Fig. 1, doing so would make it easier to follow what is happening in Fig. 1C and D. Similarly "H4 helix" is not present in Figure 1. This is very confusing as you have residues from multiple helices interacting with what I am assuming is the red IpaA helix. Please clarify this.
6. "MARCOIL predicts the presence of a coiled-coil domain in the same H10 helix of D2 between vinculin residues 348 to 393, buried into the D2 helix bundles and adjacent to the IpaA VBS3 interaction sites in the close D1D2 conformer". Are you saying the H10 Helix is not a single helix but two that interact, or that the H10 helix interacts with another helix to form a coiled-coil region?
 7. "dissociating an aVBD molecule in SEC-MALS.." spell out SEC-MALS, MALS is not defined in the text anywhere.
 8. Figure 2A legend. "The indicated stoichiometry is inferred from previous SEC-MALS analysis (Valencia-Gallardo et al., 2023) molecular and the mass of complexes" I think there is an extra word in here. Molecular?
 9. Fig. 2C "As shown in Fig. 2C, the relative amounts of D1D2 and AVBD were similar for both A and A' samples..." It is unclear to me how you would separate peak A from peak B to state that the ratios are similar. Presenting the densitometry and quantification of panel C so that the ratio can be determined would be helpful. "both values being consistent with a mixture of D1D2:aVBD 3:0 and 3:1" The molar ratio displayed in Figure 2A is not D1D2:aVBD but rather aVBD:D1D2, which makes it confusing. It would be easier to follow if they were consistently written.
 10. Figure 3: It is difficult to interpret these experiments as there is no control presented in panel A. Are the blue dots for the parent from data generated within these experiments, or a historical control/repeat of data from figure 2? If in parallel, the blots should be included in the panel A. The blue dotted lines appear to be different in each of the figures suggesting they are from different measurements, but it is unclear. Please clarify what is happening. It would be preferable that the control is done in parallel to control for the reaction conditions in each of the experiments, to know how the mutants behave relative to the control.
 11. Text states: "Type 1 mutations D389R, E143K and V367D showed no difference in D1D2 monomer disappearance, while type 2 mutations K386D, K386G and K139E showed increased rates in D1D2 monomer disappearance (Fig. 3A)." I am confused, only D389R, K386D, N393D are presented in the Figure 3. Is there supposed to be more data to examine or is this supposed to be associated with Fig. 4? It seems this paragraph in the text and the subsequent two are referring to the wrong figure.
 12. Figure 4B. What is the "CC" mutant? It is not presented in the table above in panel A or described in the text discussing the mutants or described in the legend. It is also unclear to me how from two types of mutants you are classifying them into three distinct effects on the order of the complex formation, wouldn't this imply there are three types of mutants. Could you specify what order you are referring to for the 1:0 etc means in terms of D1D2:aVBD in the figure? The discussion around this figure seems rather speculative and would likely be better placed with the material in the discussion section.
 13. Fig. 5 A, label what the panels are so we know what we are looking at.
 14. "As shown in Figure 4, the profiles of bacterial association and invasion were generally similar, suggesting consistent with a prominent role in bacterial attachment to the cell during Shigella invasion." As stated it is unclear what the authors intention is.
 15. Figure 5, western showing the levels of the vinculin in the cells would show how stable the mutants are and that, particularly for the mutations showing a phenotype, there are similar amounts of vinculin present. Since vinculin function is affected in the subsequent figure, do cells with vinculin mutants become more rounded or do not adhere such that the monolayer integrity is lost, which could confound the results? If vinculin mutants affect cell morphology such as rounding, is this finding really just a matter of flatter cells are more likely to associate and be infected with bacteria? If the vinculin level are different or the vinculin focal adhesion complexes are less pronounced, is there a difference in the amount of actin fiber formation in the cells?
 16. Fig. 7C-D The treatment timeframe is not clear. States 20 min for C in legend, but for D states 15 or 30 min with different size circles. Circles appear the same size in all conditions in D, but appear to be different sizes in C. Does this mean the treatments in C were actually for different amounts of time? It seems they should all be incubated for the same amount of time to interpret the differences between the conditions. GFP control should be presented in Fig. 7D.

Reviewer #2 (Comments to the Authors (Required)):

The authors examined the effects of certain mutations of the vinculin D1D2 subdomains that are predicted to interact with IpaA VBS3. In addition to the fundamental interests associated with these types of studies, because bacterial infection with Shigella and other microorganisms is mediated in part through vinculin and its particular structural effects on cell adhesion and internalization, the study of these vinculin mutations could have over the long-run, some potential clinical applications. In this manuscript, the focus is very much on the fundamental structural issues. The vinculin mutations that were examined in this manuscript altered the rates of D1D2 vinculin trimer formation and the measurements of disappearance of the monomer. These data are convincing and nicely relate to each other. In addition these data seem to correlate well with structures that may be indicative of a closed and open D1D2 conformer that are caused by IpaA. The paper shows that certain mutations that target the closed D1D2 conformer inhibited Shigella invasion of host cells. In contrast, mutations that targeted the open D1D2 conformer and that may affect later stages during which vinculin head-domain oligomers had markedly different effects. The authors found that all of the mutations examined in the report had an effect on the generation of focal adhesions. The authors conclude that these data indicate that vinculin is supra-activated, although this conclusion is not supported by direct assays of activation and the cell read-outs reported here are not that clear-cut. The authors go on to conclude that IpaA-induced vinculin supra-activation reinforces adhesion to matrix proteins in infected cells. The authors suggest that this process is more marked than the effect of increasing bacterial invasion. Notably, the authors did not clearly indicate in the Materials and Methods how they differentiated between surface bound and internalized bacteria. They went on to conduct shear force experiments that they interpret to indicate

that the ability of IpaA to induce vinculin supra-activation is a critical process that promotes the strengthening of cell to matrix adhesions, which they did not measure.

The manuscript is well-written but the cell-based data shown in Figures 5-7 that describes measurements of focal adhesions, cell motility and cell detachment in a microfluidics system are not consistent with the extent of the differences of oligomerization shown in the first few figures. Moreover, and as noted in the description of the results provided above, some of the measures conducted do not measure or certainly cannot provide the insights into the nature of the cell adhesion or activation processes that the authors impute. Perhaps some of these conclusions should be toned down or at least only reflect the nature of the phenomena that can truly be assessed.

One small, but possibly useful notion is that to broaden the potential audience for this paper, have the authors looked at any SNPs that may affect the open reading frames of vinculin that could affect the structure of vinculin as examined here and that therefore could alter the resistance or susceptibility of individuals to the infections that are considered here?

POINT-BY-POINT ANSWER TO THE REFEREES

Reviewer #1 (Comments to the Authors (Required)):

Cocom-Chan et. al. provide an analysis of the Shigella IpaA-vinculin interaction. The profusive use of abbreviations significantly impairs the readability and an ability to interpret the authors conclusions. The figure references are wrong throughout and made this reviewer have to guess whether the statements presented in the text were associated with particular figures. On its own the inability to read the manuscript and connect the figures to statements significantly limits the enthusiasm for the study.

1. Fig. 1A - left: shows two D2 domains and no D3 Domain. If this is correct, could you clarify what is being depicted in the legend?

We apologize for the mislabeling. The duplicate D2 domain was relabeled D3.

2. Fig. 1A - right: it is difficult to interpret what is happening. Are there coils drawn onto the structure for IpaA? It appears there is something there in green above the arrow for VBS2, but it is very difficult for me to see it clearly. I would either make these clearer as in panel B or remove them. Are the colors that are present on the left panel the same as in the right or is one of the 3D constructions supposed to represent IpaA?

We apologize for the artefactual green edging that was removed. There are no IpaA coils in Fig. 1A. The arrows in this panel indicate where the various IpaA VBS1-3 interact on full length vinculin as stated in the legend : « Right: The arrows point at binding of: IpaA VBS1 to the D1 first bundle, IpaA VBS2 to the second D1 bundle, IpaA VBS3 binding to the D1D2 interface and D2 second bundle. » The colors in the right an left panels are the same. The D1 and D2 colors have now been changed to match those of Fig. 1B.

3. The results state " Structural models indicate that aVBD triggers a 30 % angle displacement of the major axis of the D1 and D2 subdomains relative to the apo D1D2 or D1D2 in complex with IpaA VBS1-2 only (Fig. 1B;...", Yet Fig. 1B shows complex with all three VBS domains. Please clarify what is happening in the figure. What do the authors mean by aVBD in this legend. There is no aVBD on the figure. And it is not defined in the legend.

We are sorry about the omission. aVBD was defined bottom of p. 4 in the introduction but not in the legend. We have now also detailed it in the legend as follows : « B-D, cross-linking mass spectrometry-based modelling based models of 1:1 complexes of D1D2 and the IpaA domain containing all three IpaA VBS1-3 (aVBD) (Valencia-Gallardo et al, 2023).".

4. Why does the arrangement of the vinculin domains in panel A seem to not match the arrangement of the domains in panel B when the 3 domains are interacting. Please clarify what it happening. It seems to suggest that the D1-D2 interaction studies to not match the previous crystal structure of all of the domains? It appears the interactions in panel A match the domain orientation when there is only VBS1 and 2 binding, but all three VBS interactions are presented?

In Figure 1, Panel A shows full length vinculin with all D1-4 and Vt domains while Panel B only shows the D1D2 domains. The arrows in Fig.1A right point to the sites where the corresponding IpaA VBSs are depicted on D1D2 in Fig. 1B. What could have mislead the referee is that the D1 and D2 domain in Fig.1B are depicted in lighter shade colors than in in Fig 1A. The reason to this was

because Fig. 1A is derived from the crystal structure published by Bakolitsa et al, 2004, while Fig. 1B correspond to structural models derived from cross-links mass spectrometry analysis. We realized that the change in colors for the D1 and D2 domains was confusing and have now used the same colors in Fig. 1A and Fig. 1B for these subdomains.

5. "As shown in Fig. 1B, in the closed D1D2:aVBD complex, IpaA VBS3 mainly interacts with the H10 helix of D2." No H10 helix is in Fig. 1B. Moreover, there is no "H10 helix" label anywhere in Fig. 1, doing so would make it easier to follow what is happening in Fig. 1C and D. Similarly "H4 helix" is not present in Figure 1. This is very confusing as you have residues from multiple helices interacting with what I am assuming is the red IpaA helix. Please clarify this.

We are sorry, the sentence referred to Fig. 1C and not Fig. 1B. The text has been corrected. We agree with the referee : the H4 and H10 labels have been added to Fig. 1C.

6. "MARCOIL predicts the presence of a coiled-coil domain in the same H10 helix of D2 between vinculin residues 348 to 393, buried into the D2 helix bundles and adjacent to the IpaA VBS3 interaction sites in the close D1D2 conformer". Are you saying the H10 Helix is not a single helix but two that interact, or that the H10 helix interacts with another helix to form a coiled-coil region?

We are sorry about the lack of clarity. The confusion may have been generated by « bundles » that should have been singular and the adjective « same» in the sentence. H10 is a single D2 helix, that interacts with other helices in the D2 helix bundle. The sentence p. 6, l. 29 now reads : «... MARCOIL predicts the presence of a coiled-coil domain in the H10 helix of D2 between vinculin residues 348 to 393, buried into the D2 helix bundle and adjacent to the IpaA VBS3 interaction sites in the close D1D2 conformer (Delorenzi and Speed, 2002; Fig. S1).» We have added Panels C and D in Fig. S1 to show the coiled-coil domain in the ribbon structure of H10 and its masking by interaction other D2 helices in the closed and open conformers. The legend to Fig. S1 has been changed accordingly.

7. "dissociating an aVBD molecule in SEC-MALS.." spell out SEC-MALS, MALS is not defined in the text anywhere.

The abbreviation has been spelled out p. 7, l. 9 . « ... (Size Exclusion Chromatography-Multi-Angle Light Scattering)... »

8. Figure 2A legend. "The indicated stoichiometry is inferred from previous SEC-MALS analysis (Valencia-Gallardo et al., 2023) molecular and the mass of complexes" I think there is an extra word in here. Molecular?

We thank the referee for pointing at this mistake. The sentence nows reads : "The indicated stoichiometry is inferred from previous SEC-MALS analysis (Valencia-Gallardo et al., 2023) and the mass of complexes..."

9. Fig. 2C "As shown in Fig. 2C, the relative amounts if D1D2 and AVBD were similar for both A and A' samples..." It is unclear to me how you would separate peak A from peak B to state that the ratios are similar. Presenting the densitometry and quantification of panel C so that the ratio can be determined would be helpful. "both values being consistent with a mixture of D1D2:aVBD 3:0 and 3:1" The molar ratio

displayed in Figure 2A is not D1D2:aVBD but rather aVBD1D2, which makes it confusing. It would be easier to follow if they were consistently written.

We believe that we have little contamination from peak B, since the peaks A and B elute at a 1.5 mls interval and we fractionate 100 uls in the ascending phase of peak A.

The quantification of the densitometry is now presented in Figure 2.

We agree with the referee that the switch from aVBD:D1D2 and D1D2 :aVBD complexes between Figures 2A and 2C, as well the use of molar ratio for complexes and concentrations used are confusing. We are now also referring to the molar ratio of D1D2 :aVBD complexes in Figs. 2A, 2C and used the term « aVBD relative concentration » in Figs. 2B, D, E. The legend to Fig. 2 has been changed accordingly.

10. Figure 3: It is difficult to interpret these experiments as there is no control presented in panel A. Are the blue dots for the parent from data generated within these experiments, or a historical control/repeat of data from figure 2? If in parallel, the blots should be included in the panel A. The blue dotted lines appear to be different in each of the figures suggesting they are from different measurements, but it is unclear. Please clarify what is happening. It would be preferable that the control is done in parallel to control for the reaction conditions in each of the experiments, to know how the mutants behave relative to the control.

We are sorry about the lack of clarity. In Figs 2, 3 and S3, for all constructs, including the parental D1D2, the gels shown correspond to a representative experiment. The traces correspond to a linear fit of determinations in at least three independent experiments, as mentioned in the legend to the Figure. In Figure 3, the blue dots correspond to the linear fits obtained for parental D1D2 reported from Fig. 2D. This is now being mentioned in the Legend to Fig. 3. We understand the referee's concern about including a control for each set of experiments to account for possible experiment-to-experiment variation. Indeed, all experiments included the parental D1D2 as a control. At the end, we found that determinations were sufficiently reproducible to aggregate them in a single graph. We believe that the depiction of the linear fit with a dotted line could have been the source of confusion. Linear fits are now represented as plain lines.

11. Text states: "Type 1 mutations D389R, E143K and V367D showed no difference in D1D2 monomer disappearance, while type 2 mutations K386D, K386G and K139E showed increased rates in D1D2 monomer disappearance (Fig. 3A)." I am confused, only D389R, K386D, N393D are presented in the Figure 3. Is there supposed to be more data to examine or is this supposed to be associated with Fig. 4? It seems this paragraph in the text and the subsequent two are referring to the wrong figure.

We apologize for the mistake and for omitting the reference to Table 1 and Fig. S3. The text now reads : « Class 1 mutations D389R, E143K and V367D showed no difference in D1D2 monomer disappearance, while class 2 mutations K386D and K139E showed increased rates in D1D2 monomer disappearance (Table 1, Figs. 3A and S3). »

12. Figure 4B. What is the "CC" mutant? It is not presented in the table above in panel A or described in the text discussing the mutants or described in the legend. It is also unclear to me how from two types of mutants you are classifying them into three distinct effects on the order of the complex formation, wouldn't this imply there are three types of mutants. Could you specify what order you are referring to for the 1:0 etc means in terms of D1D2:aVBD in the figure? The discussion around this figure seems rather speculative and would likely be better placed with the material in the discussion section.

We are sorry about the omission. The « CC » or « cysteine clamp » mutation refers to the Q68C A396C substitutions introduced in vinculin that prevents vinculin head oligomerization but not canonical activation leading to unveiling of the vinculin tail and F-actin binding described in Valencia-Gallardo et al., 2023. We have now specified this by modifying the text:

p. 5, l.3 : « **Additionally, analysis of a vinculin cysteine-clamp variant (HV-CC), deficient for vinculin supra-activation but still proficient for canonical activation, suggests that vinculin head domain oligomerization is required for vinculin-dependent actin bundling and the maturation of focal adhesions into large adhesion structures (Valencia-Gallardo et al., 2023).** »

p. 9, l. 20 : **“The absence of effects associated with the cysteine clamp (CC) and E143K mutations...”**

p. 15, l. 15: **“The CC mutation corresponds to substitutions of glutamine 69 and Alanine 386 by cysteine residues preventing the formation of IpaA-induced vinculin oligomers but not F-actin binding upon vinculin activation (Valencia-Gallard et al., 2023).”**

We agree with the referee that there is more than three effects of mutations. Our kinetic constant analysis, however, only enabled us to distinguish two classes of mutations : 1 : mutations affecting trimer formation, but not monomer disappearance ; 2 : mutations affecting trimer formation and monomer disappearance. In Fig. 4B, we presented a speculative model of the effects of the mutations that also takes into account other considerations based on structural modeling and phenotypic effects on focal adhesions. As specified in the legend to former Fig. 4B (now Fig. S7, the 1:0, etc... correspond to the D1D2:aVBD molar ratio, with 1 :0 corresponding to apo D1D2. To distinguish between « classes » and « effects » of mutations and complying with the referee’s suggestion, in this revised version, we now present former Fig. 4B as Fig. S8 and moved the corresponding comment in the discussion. Former Fig. 4A is now referred as Table 1, former Table 1 as Table 2, and Figs. 5, 6, 7 as Figs. 4, 5, 6, respectively.

13. Fig. 5 A, label what the panels are so we know what we are looking at.

The panels were labeled.

14. "As shown in Figure 4, the profiles of bacterial association and invasion were generally similar, suggesting consistent with a prominent role in bacterial attachment to the cell during *Shigella* invasion." As stated it is unclear what the authors intention is.

We apologize for the extra word. The sentence now reads : « "As shown in Figure 4, the profiles of bacterial association and invasion were generally similar, consistent with a prominent role in bacterial attachment to the cell during *Shigella* invasion."

15. Figure 5, western showing the levels of the vinculin in the cells would show how stable the mutants are and that, particularly for the mutations showing a phenotype, there are similar amounts of vinculin present. Since vinculin function is affected in the subsequent figure, do cells with vinculin mutants become more rounded or do not adhere such that the monolayer integrity is lost, which could confound the results? If vinculin mutants affect cell morphology such as rounding, is this finding really just a matter of flatter cells are more likely to associate and be infected with bacteria? If the vinculin level are different or the vinculin focal adhesion complexes are less pronounced, is there a difference in the amount of actin fiber formation in the cells?

We have performed Western-blot analysis of cell lysates of transfectants of vinculin deficient MEF cells, showing the stability of the various vinculin-mCherry constructs. Of note, for the effects of the constructs on focal adhesions, we selected cells that expressed similar mCherry fluorescence levels. This is now indicated in the Materials and Methods p. 18, l. 13: « Cells expressing similar levels of average fluorescence intensity of the HV-mCherry constructs were analyzed. ».

Vinculin KO (Vin -/-) MEF cells have been used in many studies. While showing lower traction forces on the ECM than vinculin expressing cells, Vin -/- MEF cells do not show adhesion and drastic flattening or morphological changes in the actin cytoskeleton, with actin stress fibers traversing the cell length (for example, see Rosowski, K.A., Boltyanskiy, R., Xiang, Y. *et al.* Vinculin and the mechanical response of adherent fibroblasts to matrix deformation. *Sci Rep* 8, 17967 (2018). <https://doi.org/10.1038/s41598-018-36272-9>). As opposed to focal adhesions and consistent with previous reports, we did not observe any major differences in the actin cytoskeleton in the transfectants expressing the various vinculin-mCherry constructs.

16. Fig. 7C-D The treatment timeframe is not clear. States 20 min for C in legend, but for D states 15 or 30 min with different size circles. Circles appear the same size in all conditions in D, but appear to be different sizes in C. Does this mean the treatments in C were actually for different amounts of time? It seems they should all be incubated for the same amount of time to interpret the differences between the conditions. GFP control should be presented in Fig. 7D.

We apologize for the lack of clarity and realized that there was a mix in the presentation of the data between Panels C and D. Panel C now show the effects of time prior to shear stress on aVBD-transfected cells (Panel C). The increased adhesion under shear stress observed for aVBD-transfectants when cells are allowed to adhere for 15 min, while no effects are observed at 30 min, is consistent with aVBD accelerating endogenous processes occurring during mechanotransduction. Panel D shows the effects of vinculin depletion by siRNA and effects of the cysteine clamp vinculin variant.

The Figs 6C, D, legend and related text p. 11, last sentence- p 12, were modified accordingly. The legend to current Fig. 6 now reads : p. 31, last sentence : « Cells were perfused in a microfluidic chamber and allowed to adhere for: C, 30 min (small solid circles) and 15 min (large solid circles) prior to shear stress application; or D, 20 min.»

Reviewer #2 (Comments to the Authors (Required)):

The authors examined the effects of certain mutations of the vinculin D1D2 subdomains that are predicted to interact with IpaA VBS3. In addition to the fundamental interests associated with these types of studies, because bacterial infection with Shigella and other microorganisms is mediated in part through vinculin and its particular structural effects on cell adhesion and internalization, the study of these vinculin mutations could have over the long-run, some potential clinical applications. In this manuscript, the focus is very much on the fundamental structural issues. The vinculin mutations that were examined in this manuscript altered the rates of D1D2 vinculin trimer formation and the measurements of disappearance of the monomer. These data are convincing and nicely relate to each other. In addition these data seem to correlate well with structures that may be indicative of a closed and open D1D2 conformer that are caused by IpaA. The paper shows that certain mutations that target the closed D1D2 conformer inhibited Shigella invasion of host cells. In contrast, mutations that targeted the open D1D2 conformer and that may affect later stages during which vinculin head-domain oligomers had markedly different effects. The authors found that all of the mutations examined in the report had an

effect on the generation of focal adhesions. The authors conclude that these data indicate that vinculin is supra-activated, although this conclusion is not supported by direct assays of activation and the cell read-outs reported here are not that clear-cut. The authors go on to conclude that IpaA-induced vinculin supra-activation reinforces adhesion to matrix proteins in infected cells. The authors suggest that this process is more marked than the effect of increasing bacterial invasion. Notably, the authors did not clearly indicate in the Materials and Methods how they differentiated between surface bound and internalized bacteria. They went on to conduct shear force experiments that they interpret to indicate that the ability of IpaA to induce vinculin supra-activation is a critical process that promotes the strengthening of cell to matrix adhesions, which they did not measure.

The manuscript is well-written but the cell-based data shown in Figures 5-7 that describes measurements of focal adhesions, cell motility and cell detachment in a microfluidics system are not consistent with the extent of the differences of oligomerization shown in the first few figures. Moreover, and as noted in the description of the results provided above, some of the measures conducted do not measure or certainly cannot provide the insights into the nature of the cell adhesion or activation processes that the authors impute. Perhaps some of these conclusions should be toned down or at least only reflect the nature of the phenomena that can truly be assessed.

One small, but possibly useful notion is that to broaden the potential audience for this paper, have the authors looked at any SNPs that may affect the open reading frames of vinculin that could affect the structure of vinculin as examined here and that therefore could alter the resistance or susceptibility of individuals to the infections that are considered here?

We thank the referee for his pertinent comments. We agree that the effects of the mutations on focal adhesions cannot be fully explained by the biochemical defects observed that we characterized on purified proteins. We have toned down our statements by shortening the discussion by 20 % and by adding the following text p. 13, l. 19:

« Consistent with a role for vinculin-mediated head domain oligomerization, when transferred to full-length vinculin, we found that these mutations affecting D1D2 trimerization also affected the formation of focal adhesions. We did not find, however, a clear correlation between the type 1 and type 2 mutations expected to affect different steps of oligomerization in vitro and distinct defects on focal adhesion formation. Our analysis, however, was limited to the size or number of focal adhesions per cell. A more detailed analysis including measurements of traction forces exerted by focal adhesions could enable a better characterization of these vinculin mutations affecting head domain trimerization. Also, in cells vinculin activation is regulated by various process including phosphorylation and binding to PI 4,5 P₂ that may also impact on its oligomerization. “...

We also thank the referee for his suggestions to look into SNPs in vinculin in relation with bacterial infections. We looked into the Human Gene mutation Database but did not find SNPs corresponding to the mutations corresponding or related to the mutations that were engineered in our study. However, we also did not find any mutations related to published mutations affecting vinculin oligomerization mediated by the tail domain engineered in other studies (Chinthalapudi K, et al., J Cell Biol. 2014 ;

Shen et al., J. Biol. Chem. 2011). This may reflect the limited list of available SNPs or perhaps that such SNPs may result in embryonic lethality, as observed for vinculin-deficient mice.

May 20, 2024

RE: Life Science Alliance Manuscript #LSA-2023-02418-TR-A

Dr. Guy Tran Van Nhieu
Institute of Integrative Biology of the Cell
Cell Biology
1 avenue de la Terrasse
GIF-SUR-YVETTE, ESSONNE 91190
France

Dear Dr. Tran Van Nhieu,

Thank you for submitting your revised manuscript entitled "IpaA reveals distinct modes of vinculin activation during Shigella invasion and cell-matrix adhesion". We would be happy to publish your paper in Life Science Alliance pending final revisions necessary to meet our formatting guidelines.

- please address Reviewer 2's remaining comments
- please be sure that the authorship listing and order is correct
- please upload all figure files as individual ones, including the supplementary figure files; all figure legends should only appear in the main manuscript file
- please upload your main manuscript text as an editable doc file;
- please add ORCID ID for the corresponding author -- you should have received instructions on how to do so
- please add Keywords for your manuscript to our system
- please add the Twitter handle of your host institute/organization as well as your own or/and one of the authors in our system
- please clearly label the abstract in the manuscript file
- the contributions selected for Jacques Fattacioli and Alain Mauviel do not qualify them for authorship. Please either update the contributions in our system and the Author Contributions section of the manuscript or let us know if the authors need to be removed (and added eventually to the acknowledgments section)
- please use the [10 author names et al.] format in your references (i.e., limit the author names to the first 10)
- please add a Conflict of Interest statement to your main manuscript text
- please add your main, supplementary figure, and table legends to the main manuscript text after the references section
- please upload your Tables in editable .doc or excel format
- please check the labels for Figures S6, S7, S8 (there are 2 Figures called S5)
- The movie S1 has not been uploaded
- please add callouts for Figures 3A-C; 4A; S1A-D; S5A, C and Table 2 to your main manuscript text

FIGURE CHECKS:

- there is a vertical line after the first column in Figure 2B. Was the gel cut at that point? If so, indicate this with a vertical black line, and mention what the line is showing in the legend.
- please add sizes next to blots in Figure S6

A. FINAL FILES:

B. MANUSCRIPT ORGANIZATION AND FORMATTING:

Sincerely,

Reviewer #1 (Comments to the Authors (Required)):

The authors provide a very nice study of the role of IpaA in regulating focal adhesions through vinculin activation during Shigella infection.

The authors have adequately addressed my concerns.

Reviewer #2 (Comments to the Authors (Required)):

1) The authors have made surprisingly few and limited modifications to the manuscript, particularly in view of the very detailed and definitive requests for modifications made by reviewer #1.

2) For reviewer #2, the earlier request for clarification noted that "The authors found that all of the mutations examined in the report had an effect on the generation of focal adhesions. The authors conclude that these data indicate that vinculin is supra-activated, although this conclusion is not supported by direct assays of activation and the cell read-outs reported here are not that clear-cut." While the authors have indicated in their response to previous reviews that they have toned down their

conclusions, the cell-based data in Figures 5 and 6 don't show substantial differences between the various mutants, which calls into question the in vivo validity of their findings and makes the reader wonder what is the biological impact of their data? This is a substantial issue that has not yet been addressed by the authors.

2) The authors concluded that IpaA-induced vinculin supra-activation reinforces adhesion to matrix proteins in infected cells and they suggest that this process is more marked than the effect of increasing bacterial invasion. But the authors did not clearly indicate in the Materials and Methods how they could definitively tell the difference between surface-bound and internalized bacteria, which limit the meaning of these data. This point has also not been addressed.

Answer to Reviewer #2

1) The authors have made surprisingly few and limited modifications to the manuscript, particularly in view of the very detailed and definitive requests for modifications made by reviewer #1.

We note that reviewer 1 was satisfied with our answers to his requests.

2) For reviewer #2, the earlier request for clarification noted that "The authors found that all of the mutations examined in the report had an effect on the generation of focal adhesions. The authors conclude that these data indicate that vinculin is supra-activated, although this conclusion is not supported by direct assays of activation and the cell read-outs reported here are not that clear-cut." While the authors have indicated in their response to previous reviews that they have toned down their conclusions, the cell-based data in Figures 5 and 6 don't show substantial differences between the various mutants, which calls into question the in vivo validity of their findings and makes the reader wonder what is the biological impact of their data? This is a substantial issue that has not yet been addressed by the authors.

We respectfully disagree with reviewer #2's comment that there are no substantial differences between the various mutants in the cell-based data of Figures 5 and 6. As shown in the analysis of focal adhesions (FAs) in Fig. 5, some mutations have either effects on the size or number per cell. The K386D mutation does not have any significant effects on FA size or number per cell, while the D389R and N393D show a drastic reduction in FA size. As opposed to other mutations, the V367D shows a drastic reduction in FA number per cell, specifically associated with elongated FAs as quantified by the Aspect Ratio index in Fig. S5. In adhesion under shear stress experiments shown in Figure 6D, the CC mutant shows a three-fold decrease compare to wild-type vinculin, a value comparable to vinculin knock-down. Our statistical analysis indicates 3 to 4 stars p-values attesting the robustness of these differences. As requested from reviewer 2's suggestion on the original submission, we have toned down our conclusions and removed speculative parts of the discussion.

2) The authors concluded that IpaA-induced vinculin supra-activation reinforces adhesion to matrix proteins in infected cells and they suggest that this process is more marked than the effect of increasing bacterial invasion. But the authors did not clearly indicate in the Materials and Methods how they could definitively tell the difference between surface-bound and internalized bacteria, which limit the meaning of these data. This point has also not been addressed.

We apologize for the lack of clarity. We have amended the corresponding text in the Materials and Methods section as follows :

“Bacterial association and invasion assays

GFP-expressing bacteria were inoculated from pre-cultures at a dilution of 1:100 in trypticase soy broth containing chloramphenicol at 15 µg/ml and grown in a shaking incubator at 37°C

until reaching an optical density at 600 nm ($OD_{600\text{ nm}}$) of 0.4. Bacterial cultures were centrifuged at 7500 g for 10 min and the bacterial pellet was resuspended in EM buffer (120 mM NaCl, 7 mM KCl, 1.8 mM $CaCl_2$, 0.8 mM $MgCl_2$, 5 mM glucose, and 25 mM HEPES, pH = 7.3) at a final $OD_{600\text{ nm}}$ of 0.1. Cells were incubated with the bacterial suspension for 60 min at 37°C prior to fixing and processing for fluorescence microscopy analysis. Samples were fixed in PBS containing 3.7 % paraformaldehyde for 645 min at 21°C, prior to processing for fluorescence staining of F-actin using Phalloidin-Alexa 488. Extracellular and internalized bacteria were scored using an “inside-out” staining procedure as previously described (Valencia-Gallardo et al., 2023). Briefly, samples were incubated for 45 min at room temperature with a rabbit anti-LPS antibody diluted 1:5000 in PBS in the absence of permeabilization, followed incubation with an anti-rabbit IgG antibody coupled to Alexa Fluor 635 diluted 1:200 to label extracellular bacteria. Samples were then permeabilized by incubation with PBS containing 0.1% Triton X-100 for 4 min, prior to incubation with Phalloidin-Alexa 350 diluted 1:5000 to label F-actin. Samples were mounted in DAKO mounting media. Bacterial association with cells was determined by scoring all GFP-expressing bacteria. Bacterial invasion was determined by scoring GFP-expressing bacteria that did not label with Alexa Fluor 650. Bacterial counts were normalized to the cell number.”

May 24, 2024

RE: Life Science Alliance Manuscript #LSA-2023-02418-TRR

Dr. Guy Tran Van Nhieu
Institute of Integrative Biology of the Cell
Cell Biology
1 avenue de la Terrasse
GIF-SUR-YVETTE, ESSONNE 91190
France

Dear Dr. Tran Van Nhieu,

Thank you for submitting your Research Article entitled "IpaA reveals distinct modes of vinculin activation during Shigella invasion and cell-matrix adhesion". It is a pleasure to let you know that your manuscript is now accepted for publication in Life Science Alliance. Congratulations on this interesting work.

DISTRIBUTION OF MATERIALS:

Again, congratulations on a very nice paper. I hope you found the review process to be constructive and are pleased with how the manuscript was handled editorially. We look forward to future exciting submissions from your lab.

Sincerely,
